# Meta Optimality for Demographic Parity Constrained Regression via Post-Processing

**Kazuto Fukuchi** [1] [2]

## Abstract

We address the regression problem under the constraint of demographic parity, a commonly used fairness definition. Recent studies have revealed fair minimax optimal regression algorithms, the most accurate algorithms that adhere to the fairness constraint. However, these analyses are tightly coupled with specific data generation models. In this paper, we provide meta-theorems that can be applied to various situations to validate the fair minimax optimality of the corresponding regression algorithms. Furthermore, we demonstrate that fair minimax optimal regression can be achieved through post-processing methods, allowing researchers and practitioners to focus on improving conventional regression techniques, which can then be efficiently adapted for fair regression.

## 1. Introduction

Machine learning systems have become increasingly prevalent in decision-making across various domains, including healthcare, finance, and criminal justice. While these systems promise more efficient and data-driven decisions, they also raise significant concerns regarding fairness and equity. As machine learning models learn from historical data, they can inadvertently perpetuate or even exacerbate existing societal biases, leading to unfair outcomes for certain social groups. Numerous instances of unfair behaviors in real-world machine learning systems have been reported, including biased recidivism risk prediction (Angwin et al., 2016), discriminatory hiring practices (Dastin, 2018), inequitable facial recognition performance (Crockford, 2020; Najibi, 2020), and biased credit scoring (Vigdor, 2019). These reports underscore the urgent need to address the issue of unfairness in machine learning.

To mitigate unfair bias, researchers have developed diverse methodologies for constructing accurate predictors that ensure certain fairness definitions. These methodologies can be categorized into three main types: pre-processing techniques that modify the training data (Feldman et al., 2015), in-processing methods that incorporate fairness constraints during model training (Chuang & Mroueh, 2020; Zhang et al., 2018; Du et al., 2021; Cotter et al., 2019; Khalili et al., 2023; Jovanović et al., 2023), and post-processing approaches that adjust the model's outputs (Menon & Williamson, 2018; Chzhen et al., 2019; Jiang et al., 2020; Schreuder & Chzhen, 2021; Chen et al., 2023; Xu & Strohmer, 2023; Xian et al., 2023; Zhao & Gordon, 2019; Chzhen et al., 2020). Methodologies of each type have been proposed for various fairness definitions, including demographic parity (Pedreshi et al., 2008), equalized odds (Hardt et al., 2016), multicalibration (Kleinberg et al., 2017), and individual fairness (Dwork et al., 2012). Each of these fairness criteria aims to address different aspects of algorithmic bias. These diverse methodologies have significantly advanced the field of fair machine learning, enabling researchers and practitioners to address fairness concerns in various contexts and applications.

Recent advancements in fair learning algorithms have led to the development of methods that achieve the best possible predictive accuracy while adhering to specific fairness definitions, particularly demographic parity. Such advancements were achieved by constructing fair learning algorithms and proving their minimax optimality under regression (Chzhen & Schreuder, 2022; Fukuchi & Sakuma, 2023) and classification (Zeng et al., 2024) setups. The fair minimax optimal algorithm is an algorithm that satisfies the fairness definition and minimizes the worst-case error taken over a certain set of data generation models. No fair algorithm can outperform the fair minimax optimal algorithm in the sense of the worst-case error, as the minimization is taken over all the fair algorithms.

While these optimal methods represent significant progress, they are often tightly coupled with specific data generation models, limiting their applicability to a broader range of real-world scenarios. For example, Chzhen & Schreuder

---

[1] Institute of Systems and Information Engineering, University of Tsukuba, Ibaraki, Japan [2] RIKEN AIP, Tokyo, Japan. Correspondence to: Kazuto Fukuchi <fukuchi@cs.tsukuba.ac.jp>.

*Proceedings of the 42nd International Conference on Machine Learning*, Vancouver, Canada. PMLR 267, 2025. Copyright 2025 by the author(s).

(2022) and Fukuchi & Sakuma (2023) each employ certain linear models of the outcome with Gaussian features, albeit with different specific formulations. Zeng et al. (2024) work under assumptions including that the regression function is within the Hölder class and that both the margin and strong density conditions are satisfied. The dependence on these particular data distributions and model assumptions can restrict the applicability of these approaches, potentially hindering their adoption in diverse applications where the underlying data characteristics may differ significantly from these specific conditions.

**Our contributions** (**Meta-optimality**) We address the limitations of existing analyses for fair regression by establishing a meta-theorem that applies to a wide range of scenarios. This meta-theorem provides a connection between the minimax optimal error for fair regression and that for conventional regression, allowing for rates tailored to various situations by leveraging well-established results regarding the minimax optimality in conventional regression. Our approach can combine with minimax optimal regressions under diverse smoothness assumptions (e.g., Hölder, Sobolev, and Besov spaces (Donoho & Johnstone, 1998; Giné & Nickl, 2015)) and minimax optimal deep learning methods (Schmidt-Hieber, 2020; Suzuki, 2018; Suzuki & Nitanda, 2021; Nishimura & Suzuki, 2023).

(**Optimal fair regression by post-processing**) We propose a post-processing algorithm that leverages an optimal conventional regression algorithm. Guided by our meta-theorem, this construction ensures minimax optimality under the assumptions employed by the conventional regression algorithm. Since the proposed algorithm is post-processing, practitioners can concentrate on refining conventional regression methods, which can then be seamlessly adapted for fair regression.

(**Convergence rate analysis for optimal transport map estimation in Wasserstein barycenters**) A key component of our algorithm is optimal transport map estimation within the Wasserstein barycenter problem, which seeks a distribution (often called the barycenter) that minimizes the (Wasserstein) distances to a set of distributions. The optimal transport map is a mapping between distributions that achieves the minimum cost. One of our main contributions is to provide a convergence rate analysis of a transport map estimator for the Wasserstein barycenter problem, which may be of independent interest. The analyzed estimator is based on Korotin et al. (2020), but they did not provide a convergence rate analysis. The detailed discussion will appear in Section 7.

All the missing proofs can be found in Appendix A.

**Notations** For a positive integer $m$, let $[m] = \{1, ..., m\}$ and $\Delta_m$ denote the probability simplex over $[m]$. We denote the indicator function by $\mathbb{1}$. For real values $a$ and $b$, we define $a \vee b = \max\{a, b\}$ and $a \wedge b = \min\{a, b\}$. For a sequence $a_t$ indexed by $t \in \mathcal{T}$, we represent the family $(a_t)_{t \in \mathcal{T}}$ as $a_:$. The first derivative of a function $f : \mathbb{R} \to \mathbb{R}$ is denoted by $Df$. Given an event $\mathcal{E} \in \mathfrak{Z}$ in a probability space $(\mathcal{Z}, \mathfrak{Z}, \nu)$, we denote its complement by $\mathcal{E}^c$ and its probability by $\mathbb{P}_\nu\{\mathcal{E}\}$. For a random variable $X$ from $(\mathcal{Z}, \mathfrak{Z}, \nu)$ to a measurable space $(\mathcal{X}, \mathfrak{X})$, we denote its expectation and variance by $\mathbb{E}_\nu[X]$ and $\mathbb{V}_\nu[X]$, respectively.

## 2. Problem Setup and Preliminaries

### 2.1. Fair Regression Problems

Consider a fair regression problem with $M \geq 2$ social groups. Let $\mathcal{X}$ and $\Omega \subset \mathbb{R}$ be the domains of features and outcomes, respectively, where we assume $\Omega$ is open and bounded. For each social group $s \in [M]$ (e.g., male and female for gender), let $X^{(s)} \in \mathcal{X}$ and $Y^{(s)} \in \Omega$ be random variables on a probability measure space $(\mathcal{Z}, \mathfrak{Z}, \mu_s)$, representing the features and outcomes of an individual in group $s$, respectively. The goal of the regression problem is to construct a (group-wise) regressor $f_:$, mappings from $\mathcal{X}$ to $\Omega$ indexed by $s \in [M]$, that accurately predicts $Y^{(s)}$ based on $X^{(s)}$. The ideal regressor, known as the Bayes-optimal regressor, is defined as $f^*_{\mu,s} = \arg\min_f \mathbb{E}_{\mu_s}[(f(X^{(s)}) - Y^{(s)})^2]$ and is given by $f^*_{\mu,s}(X^{(s)}) = \mathbb{E}_{\mu_s}[Y^{(s)} \mid X^{(s)}]$. We use $\mu_{Y,s}$ and $\mu_{X,s}$ to denote the laws of $Y^{(s)}$ and $X^{(s)}$, respectively. Additionally, we denote the law of $f^*_{\mu,s}(X^{(s)})$ by $\mu_{f,s}$.

Given samples consisting of $n_s$ i.i.d. copies of $(X^{(s)}, Y^{(s)})$, the objective of the learning algorithm is to construct a regressor $\bar{f}_{n,:}$ that maximizes accuracy while satisfying a fairness constraint. Let $n = \sum_{s \in [M]} n_s$ for notational convenience. We now introduce the definition of fairness, define a measure of accuracy, and provide the definition of the fair optimal algorithm.

**Fairness** We employ demographic parity (Pedreshi et al., 2008) as our fairness criterion. A regressor $f_:$ satisfies demographic parity if its output distribution remains invariant across all groups $s \in [M]$.

**Definition 2.1.** A regressor $f_:$ satisfies (strict) demographic parity if, for all $s, s' \in [M]$ and for all events $E$, $\mathbb{P}_{\mu_s}\{f_s(X^{(s)}) \in E\} = \mathbb{P}_{\mu_{s'}}\{f_{s'}(X^{(s')}) \in E\}$.

Let $\bar{\mathcal{F}}(\mu_{X,:})$ denote the set of all regressors satisfying demographic parity for given laws $\mu_{X,:}$.

Instead of enforcing strict demographic parity in Definition 2.1, we adopt the concept of *fairness consistency* (Chzhen et al., 2020; Fukuchi & Sakuma, 2023).

**Definition 2.2.** A learning algorithm is consistently fair if $\bar{f}_{n,:}$ converges in probability to an element of $\bar{\mathcal{F}}(\mu_{X,:})$ as $n_1, ..., n_M$ approach infinity.

Definition 2.2 implies that a consistently fair learning algorithm eventually constructs a regressor that satisfies strict demographic parity given a sufficiently large sample size.

**Accuracy** We evaluate the accuracy of a given regressor by measuring its expected squared distance from the fair Bayes-optimal regressor. Given probability measures $\nu_:$ on a measurable space $(\mathcal{Z}, \mathfrak{z})$ indexed by $[M]$ and weights $w_: \in \Delta_M$ (known to the learner), we define the squared distance $d_{\nu_:}$ between functions $f_s, f'_s : \mathcal{Z} \to \mathbb{R}$ indexed by $s \in [M]$ as

$$d^2_{\nu_:}(f_:, f'_:) = \sum_{s \in [M]} w_s \int (f_s(z) - f'_s(z))^2 \mu_s(dz)$$
$$:= \sum_{s \in [M]} w_s d^2_{\nu_s}(f_s, f'_s).$$

The fair Bayes-optimal regressor is defined as the regressor that satisfies strict demographic parity and minimizes the deviation from the Bayes-optimal regressor:

$$\bar{f}^*_{\mu,:} = \underset{f_: \in \bar{\mathcal{F}}(\mu_{X,:})}{\arg\min}\ d^2_{\mu_{X,:}}(f_:, f^*_{\mu,:}).$$

The accuracy of a regressor $f_:$ is then evaluated by $d^2_{\mu_{X,:}}(f_:, \bar{f}^*_{\mu,:})$.

**Optimality** The fair minimax optimal algorithm for a set of distributions $\mathcal{P}$ is a consistently fair algorithm that achieves the fair minimax optimal error over $\mathcal{P}$. The fair minimax optimal error over $\mathcal{P}$ is defined as

$$\bar{\mathcal{E}}_n(\mathcal{P}) = \inf_{\bar{f}_{n,:}: \text{fair}} \sup_{\mu_: \in \mathcal{P}} \mathbb{E}_{\mu^n_:}[d^2_{\mu_{X,:}}(\bar{f}_{n,:}, \bar{f}^*_{\mu,:})], \qquad (1)$$

where the infimum is taken over all consistently fair learning algorithms, and $\mathbb{E}_{\mu^n_:}$ denotes the expectation over samples. Thus, no consistently fair learning algorithm can outperform the fair minimax optimal algorithm in terms of worst-case expected deviation.

### 2.2. Fair Bayes-Optimal Regressors and Optimal Transport Maps

Recent analyses have characterized fair Bayes-optimal regressors using optimal transport maps that arise in the Wasserstein barycenter problem (Chzhen et al., 2020; Chzhen & Schreuder, 2022). Given two probability measures $\nu$ and $\nu'$, the optimal transport map with a quadratic cost function is the unique solution of Monge's formulation of the optimal transportation problem between $\nu$ and $\nu'$, i.e.,

a transport map $\vartheta^* : \mathbb{R} \to \mathbb{R}$ that realizes the infimum

$$W_2^2(\nu, \nu') = \inf_{\vartheta: \vartheta \sharp \nu = \nu'} \int \frac{1}{2}(z - \vartheta(z))^2 \nu(dz),$$

where $\vartheta \sharp \nu$ denotes the pushforward measure of $\nu$ by $\vartheta$ [1]. Given probability measures $\nu_1, \ldots, \nu_k$, the Wasserstein barycenter problem with weights $w_: \in \Delta_k$ is defined as

$$\inf_\nu \sum_{i \in [k]} w_s W_2^2(\nu_i, \nu). \qquad (2)$$

We refer to the unique solution of Equation (2) as the barycenter of $\nu_:$ with weights $w_:$. The optimal transport map from $\nu_i$ to the barycenter of $\nu_:$ is denoted by $\vartheta^*_{\nu,i}$. Building on these concepts, the fair Bayes-optimal regressor is obtained as follows:

**Theorem 2.3** (Chzhen et al. (2020)). *Assume that $\mu_{f,s}$ admits a density for all $s \in [M]$. Then, the fair Bayes-optimal regressor is given by*

$$\bar{f}^*_{\mu,s}(x) = (\vartheta^*_{\mu_f,s} \circ f^*_{\mu,s})(x).$$

Theorem 2.3 reveals that the fair Bayes-optimal regressor is characterized by the Bayes-optimal regressor $f^*_{\mu,:}$ and the optimal transport maps $\vartheta^*_{\mu_f,:}$. Throughout the paper, we assume $\mu_{f,s}$ admits a density for all $\mu_: \in \mathcal{P}$ and $s \in [M]$ so that the condition of Theorem 2.3 holds.

### 2.3. Potential Minimization and Optimal Transport Maps

The transport maps in Wasserstein barycenter problems are characterized by the minimizer of *multiple correlation* over *congruent* potentials (Korotin et al., 2020). Congruent potentials with weights $w_: \in \Delta_k$ are convex and lower semi-continuous functions $u : \mathbb{R} \to \mathbb{R}$ such that $\sum_{i \in [k]} w_i Du^\dagger(z) = z$ for all $z \in \Omega$, where $u^\dagger(z) = \sup_x(zx - u(x))$ is the convex conjugate of $u$. Given probability measures $\nu_:$, the multiple correlation of $\nu_:$ with weights $w_:$ for congruent potentials $u_:$ is defined as

$$C(u_:; \nu_:) = \sum_{i \in [k]} w_i \int u_i d\nu_i.$$

Let $u^*_:$ denote the optimal congruent potentials that minimize $C(u_:; \nu_:)$. Through the analyses by Agueh & Carlier (2011); Álvarez Esteban et al. (2016), we obtain the following characterization of the optimal transport maps:

**Corollary 2.4** (Agueh & Carlier (2011); Álvarez Esteban et al. (2016)). *Let $\vartheta^*_:$ be the optimal transport maps from $\nu_:$ to the barycenter of $\nu_:$ with weights $w_:$. Suppose that for all $i \in [k]$, $\nu_i$ admits densities. Then, $\vartheta^*_i = Du^*_i$ for $i \in [k]$.*

---

[1] We use the notation $W_2$ because $W_2$ is known as the 2-Wasserstein distance.

By Corollary 2.4, we can obtain the optimal transport maps $\vartheta^*_{:}$ by solving $\inf_{u_{:}:\text{congruent}} C(u_{:}; \nu_{:})$.

# 3. Main Result

Our main result is a meta-theorem that characterizes the fair minimax optimal error (see Equation (1)), showing how its convergence rate depends on $\mathcal{P}$. We begin by introducing several technical assumptions on $\mathcal{P}$ before presenting our main meta-theorem.

For convenience, we introduce several notations. We use notations $\mathcal{P}_Y = \{\mu_{Y,:} : \mu_{:} \in \mathcal{P}\}$, $\mathcal{P}_X = \{\mu_{X,:} : \mu_{:} \in \mathcal{P}\}$, and $\mathcal{P}_f = \{\mu_{f,:} : \mu_{:} \in \mathcal{P}\}$. We denote $\mathcal{P}_s = \{\mu_s : \mu_{:} \in \mathcal{P}\}$ for $s \in [M]$. Let $\mathcal{F}_{\mathcal{P}} = \{f^*_{\mu_{:}} : \mu_{:} \in \mathcal{P}\}$, and let $\mathcal{F}_{\mathcal{P},s} = \{f^*_{\mu,s} : \mu_{:} \in \mathcal{P}\}$ for $s \in [M]$. Let $\Theta_{\mathcal{P}} = \{\vartheta^*_{\mu_{f,:}} : \mu_{:} \in \mathcal{P}\}$. We omit $\mathcal{P}$ in the subscript of these notations if $\mathcal{P}$ is clear from the context.

**Assumptions**  We begin with our first assumption.

**Assumption 3.1.**  $\mathcal{P}$ satisfies the following three conditions:

1. $\mathcal{P}_Y \times \mathcal{P}_X \subseteq \{(\mu_{Y,:}, \mu_{X,:}) : \mu_{:} \in \mathcal{P}\}$,
2. for any permutation $\pi$ over $[M]$, $\mu_{\pi(:)} \in \mathcal{P}$ if $\mu_{:} \in \mathcal{P}$,
3. $\mathcal{F}_s$ is convex, meaning for any $f, f' \in \mathcal{F}_s$ and $t \in (0,1)$, $tf + (1-t)f' \in \mathcal{F}_s$.

Intuitively, the first and second conditions imply that the learner has no prior knowledge about (i) how the distributions of $Y^{(s)}$ and $X^{(s)}$ are related, nor (ii) how the distributions $\{\mu_s\}_{s \in [M]}$ differ. These conditions are naturally satisfied in many real-world scenarios. Note that the third condition does not require the regression functions themselves to be convex but rather that the set $\mathcal{F}_s$ is convex, which can still accommodate non-convex functions.

Next, we impose assumptions on $\mu_{f,:}$ to facilitate the estimation of the optimal transport maps $\vartheta^*_{\mu_{f,:}}$. First, we assume that $\vartheta^*_{\mu_{f,:}}$ are elements of Lipschitz and strictly increasing functions $\mathcal{M}_L$ [2]. Specifically, $\mathcal{M}_L$ is defined as the set of functions $\vartheta : \Omega \to \Omega$ satisfying

$$L^{-1}(y-x) \le \vartheta(x) - \vartheta(y) \le L(y-x) \quad \forall x > y. \tag{3}$$

Second, we assume that $\mu_{f,s}$ satisfies the Poincaré-type inequality: there exists a constant $C_P > 0$ such that for any function $g : \Omega \to \Omega$ with an $L$-Lipschitz continuous gradient,

$$\mathbb{V}_{\mu_s}\left[g\left(X^{(s)}\right)\right] \le C_P \mathbb{E}_{\mu_s}\left[\left(Dg\left(X^{(s)}\right)\right)^2\right]. \tag{4}$$

**Assumption 3.2.**  There exists a constant $L > 1$ such that for all $\mu_{:} \in \mathcal{P}$ and all $s \in [M]$, 1) $\vartheta^*_{\mu_{f,s}} \in \mathcal{M}_L$ and 2) $\mu_{f,s}$ satisfies the Poincaré-type inequality in Equation (4).

[2]It is worth noting that any transport map over the real line is a non-decreasing function due to the convexity of the potential.

Note that similar assumptions to Assumption 3.2 are also employed in studies on transport map estimation, including Hütter & Rigollet (2021); Divol et al. (2024).

We use the following complexity measure of the class of transport maps $\Theta$ based on metric entropy. Given $\epsilon > 0$, the $\epsilon$-covering number of a set $A \subseteq \mathcal{X}$ with a metric space $(\mathcal{X}, d)$ is denoted as $N(\epsilon, A, d)$; namely, $N(\epsilon, A, d)$ denotes the minimum number of balls whose union covers $A$. Let $\ln_+(x) = 0 \vee \ln(x)$. The complexity measure is defined as follows:

**Definition 3.3.**  The complexity of $\mathcal{P}_f$ is $(\alpha, \beta)$ for $\alpha > 0$ and $\beta \ge 0$ if there exist constants $C, C' > 0$ and $\bar{\epsilon} > 0$, and a sequence of subsets $\Theta_1 \subseteq \Theta_2 \subseteq ... \subseteq \{\vartheta^*_{\nu,:} : \nu_{:} \in \mathcal{P}_f\}$ such that for any integer $j \ge 0$,

1. $\sup_{\nu_{:} \in \mathcal{P}_f} \inf_{\vartheta_{:} \in \Theta_j} d^2_{\nu_{:}}(\vartheta_{:}, \vartheta^*_{\nu,:}) \le C 2^{-\alpha j}$,
2. $\sup_{\nu_{:} \in \mathcal{P}_f} \ln N(\epsilon, \Theta_j, d_{\nu_{:}}) \le C' 2^{\beta j} \ln_+(1/\epsilon)$ for $\epsilon \in (0, \bar{\epsilon}]$.

In our main theorem, we will assume the complexity is $(\alpha, \beta)$ for some $\alpha$ and $\beta$.

**Meta-theorem**  We now characterize the fair minimax optimal error $\bar{\mathcal{E}}_n(\mathcal{P})$ in terms of the conventional minimax optimal error. Specifically, for a given group $s \in [M]$, the group $s$'s conventional minimax optimal error is defined as

$$\mathcal{E}_k(\mathcal{P}_s) = \inf_{f_k} \sup_{\mu_s \in \mathcal{P}_s} \mathbb{E}_{\mu^k_s}\left[d^2_{\mu_{X,s}}(f_k, f^*_{\mu,s})\right]$$

where the infimum is taken over all regression algorithms that take $k$ i.i.d. copies of $(X^{(s)}, Y^{(s)})$ as the observed sample. Let $\tilde{n} = \min_{s \in [M]} n_s / w_s$.

**Theorem 3.4.**  *Assume Assumptions 3.1 and 3.2 and that the complexity of $\mathcal{P}_f$ is $(\alpha, \beta)$ for some $\alpha > 0$ and $\beta \ge 0$. Then, there exists a consistently fair learning algorithm such that*

$$\mathcal{E}_n(\mathcal{P}_1) \le \bar{\mathcal{E}}_n(\mathcal{P}) \le C \cdot$$
$$\left(L^2 \sum_{s \in [M]} w_s \mathcal{E}_{n_s}(\mathcal{P}_s) + \left(\frac{\tilde{n}}{\ln(\tilde{n})}\right)^{-\alpha/(\alpha+\beta)}\right),$$

*for some constant $C > 0$.*

We highlight several implications of Theorem 3.4:

1. Assuming there exists a constant $c > 0$ such that $n_s \ge cw_s n$ for all $s \in [M]$, Theorem 3.4 shows that the optimal rate with respect to $n$ is $\mathcal{E}_n(\mathcal{P}_1)$ (note that $\mathcal{E}_n(\mathcal{P}_1) = ... = \mathcal{E}_n(\mathcal{P}_M)$) whenever $\mathcal{E}_n(\mathcal{P}_1)$ is larger than $\left(\frac{n}{\ln(n)}\right)^{-\alpha/(\alpha+\beta)}$. In such cases, the rate of $\bar{\mathcal{E}}_n(\mathcal{P})$ can vary with $\mathcal{P}$ along with $\mathcal{E}_n(\mathcal{P}_1)$.
2. In general, Theorem 3.4 implies that if the conventional regression problem is more difficult than the transport map estimation problem, then the optimal fair regression

error is dominated by the conventional minimax error. This situation commonly arises in high-dimensional settings for $\mathcal{X}$ (e.g., image, text, or audio regression).

3. Chzhen et al. (2020) also established a similar upper bound on the error under the demographic parity constraint. However, their result has two notable limitations. First, their analysis requires stronger assumptions than ours. For instance, they assume that the conventional regression algorithm admits a sub-Gaussian high-probability error bound, whereas our results only require a bound on the expected squared error. Furthermore, while they require uniform upper and lower bounds on the density of $\mu_{f,s}$, we only assume the Poincaré-type inequality. These differences broaden the applicability of our meta-theorem relative to their findings.

4. The second limitation of Chzhen et al. (2020) is that their results cannot achieve a convergence rate faster than $n^{-1/2}$, since their upper bound on the estimation error of $\vartheta_{\mu,:}^*$ is $n^{-1/2}$ and dominates the other terms. In contrast, our result can achieve a rate faster than $n^{-1/2}$ by exploiting the smoothness structure of $\vartheta_{\mu,:}^*$.

**Illustrative Example**   To concretely demonstrate the implications of our theoretical results, we consider a representative scenario in which the regression function $f_{\mu,s}^*$ is a composition of multiple functions, as studied by Schmidt-Hieber (2020), and the optimal transport map $\vartheta_{\mu,s}^*$ lies within a Sobolev function class. Specifically, let $f_{\mu,s}^*$ belong to the class

$$\Big\{ g_q \circ \dots \circ g_0 :$$
$$g_i = (g_{ij})_j : [a_i, b_i]^{d_i} \to [a_{i+1}, b_{i+1}]^{d_{i+1}},$$
$$g_{ij} \in C_{t_i}^{\beta_i}\big([a_i, b_i]^{t_i}\big)\Big\},$$

where $C_r^\beta$ denotes the Hölder class of functions with smoothness parameter $\beta$ and $r$-dimensional input, $d_i$ is the input dimension of $g_i$, and $t_i < d_i$ indicates that each $g_{ij}$ depends on only $t_i$ out of $d_i$ variables. This structure captures the notion of compositional functions with sparse dependencies, which is prevalent in high-dimensional statistical learning.

For regression functions of this form, the minimax optimal error for group $s$ satisfies $\mathcal{E}_n(\mathcal{P}_s) = \Theta\big(\max_i n_s^{-\beta_i^*/(2\beta_i^*+t_i)}\big)$ up to logarithmic factors, where $\beta_i^* = \beta_i \prod_{\ell=i+1}^q (\beta_\ell \wedge 1)$. Also, for the class of transport maps $\vartheta_{\mu,s}^*$ taken to be the Sobolev class of smoothness $\gamma > 0$, Definition 3.3 is satisfied with $\alpha = 2\gamma$ and $\beta = 1$ by choosing $\Theta_j$ as the span of the first $j$ wavelet basis functions. Consequently, the fair minimax error is obtained as

$$\bar{\mathcal{E}}_n(\mathcal{P}) = \Theta\Big(\max_i n^{-\beta_i^*/(2\beta_i^*+t_i)} + n^{-2\gamma/(2\gamma+1)}\Big),$$

again up to logarithmic factors, provided that there exists a constant $c > 0$ such that $n_s \geq c w_s n$ for all $s \in [M]$.

An important insight from this example is that if the smoothness of the regression components and the transport maps are comparable (i.e., $\beta_i^* \approx \gamma$), then the minimax error is dominated by the regression term whenever $t_i > 1$ for some $i$. Here, $t_i$ reflects the intrinsic dimensionality of the essential intermediate data representation. In practical situations, the intermediate representations may be multi-dimensional ($t_i > 1$), and thus the overall rate is determined by the conventional regression problem.

# 4. Optimal Algorithm and Upper Bound

In this section, we present our fair regression algorithm. Its core structure is similar to algorithms proposed by Chzhen et al. (2020); Chzhen & Schreuder (2022). Leveraging Theorem 2.3, we obtain the fair Bayes-optimal regressor by composing the optimal transport maps $\vartheta_{\mu_f,:}^*$ from $\mu_{f,:}$ to the barycenter of $\mu_{f,:}$ with the conventional Bayes-optimal regressor $f_{\mu,:}^*$. Let $f_{n,:}$ and $\vartheta_{n,:}$ be estimators of $f_{\mu,:}^*$ and $\vartheta_{\mu_f,s}^*$, respectively. We define the estimator of $\bar{f}_{\mu,:}^*$ as $\bar{f}_{n,s}(x) = (\vartheta_{n,s} \circ f_{n,s})(x)$. This procedure can be viewed as post-processing since we first construct $f_{n,:}$ using a conventional learning algorithm and then refine its outputs using $\vartheta_{n,:}$.

We employ a minimax-optimal conventional regression algorithm for the model $\mathcal{P}$ as the estimator $f_{n,:}$. Our main methodological contribution, distinguishing our work from earlier approaches, is the introduction of a suitable estimator for $\vartheta_{n,:}$. To estimate $\vartheta_{\mu_f,:}^*$, we utilize the strategy proposed by Korotin et al. (2020) (see Section 2.3). We also provide a novel analysis of the convergence rate for the estimation error of $\vartheta_{n,:}$, which was not addressed in Korotin et al. (2020).

We first describe the construction of $\vartheta_{n,:}$ and then demonstrate the overall procedure of our fair regression algorithm along with analyses of its accuracy and fairness.

**Barycenter estimation**   As discussed in Section 2.3, one can obtain the optimal transport maps in the Wasserstein barycenter problem by finding congruent potentials that minimize the multiple correlation $C(u_:, \nu_:)$. In the fair regression setting, our goal is to estimate the transport maps in the Wasserstein barycenter problem for $\mu_{f,:}$. However, the learner cannot directly observe $\mu_{f,:}$ since neither $f_{\mu,:}^*$ nor $\mu_{X,:}$ is known. Instead, we substitute $f_{n,:}$ for $f_{\mu,:}^*$ and use empirical measures from the observed samples in place of $\mu_{X,:}$. Specifically, let $\mu_{\hat{f},s}$ be the law of $f_{n,s}(X^{(s)})$ for $s \in [M]$, and let $\mu_{n,\hat{f},s}$ be the corresponding empirical measure induced by a sample of size $n_s$. Then, the estimator

**Algorithm 1** Optimal fair regression

---

**input** Samples $(Y_1^{(s)}, X_1^{(s)}), ..., (Y_{2n}^{(s)}, X_{2n_s}^{(s)})$

**output** $\bar{f}_{n,:}$

  1: Construct $f_{n,:}$ using a minimax optimal conventional regression algorithm with halves of samples $(Y_1^{(s)}, X_1^{(s)}), ..., (Y_{n_s}^{(s)}, X_{n_s}^{(s)})$

  2: Construct $\vartheta_{n,:}$ by Equation (5) with the remaining samples $f_{n,s}(X_{n+1}^{(s)}), ..., f_{n,s}(X_{2n_s}^{(s)})$

  3: $\bar{f}_{n,s}(x) = (\vartheta_{n,s} \circ f_{n,s})(x)$

---

$\vartheta_{n,:}$ is defined as the minimizer of

$$\inf_{\vartheta_: \in \Theta_j} C(u_{\vartheta,:}, \mu_{n,\hat{f},:}), \tag{5}$$

where $u_{\vartheta,:}$ denotes the potential functions corresponding to the transport maps $\vartheta_:$, and $\Theta_j$ is a sequence of subsets of $\Theta$ described in Definition 3.3.

*Remark* 4.1. We require a specific construction of $u_{\vartheta,:}$ for technical reasons. We extend the input and output domains of any function $\vartheta : \Omega \to \Omega$ to $\mathbb{R}$ while preserving the property in Equation (3) by defining $\vartheta(z) = z + c_{\sup}$ for large $z$ and $\vartheta(z) = z + c_{\inf}$ for small $z$ with appropriate constants $c_{\sup}, c_{\inf} \in \mathbb{R}$. We interpret functions in $\Theta$ as their extended versions. Letting $u_{\vartheta,s}^{\dagger}(z) = \int_0^z \vartheta_s^{-1}(x)dx$, we define $u_{\vartheta,s}$ as the convex conjugate of $u_{\vartheta,s}^{\dagger}$; i.e., $u_{\vartheta,s}(z) = \sup_{x \in \mathbb{R}}(xz - u_{\vartheta,s}^{\dagger}(x))$.

**Overall algorithm** Algorithm 1 summarizes the overall procedure of our fair regression algorithm. For simplicity, we assume that the sample size for group $s$ is $2n_s$. In the first step, we execute a minimax-optimal conventional regression algorithm with half of the samples to obtain $f_{n,:}$. By definition, $f_{n,s}$ achieves an error of $\mathcal{E}_n(\mathcal{P}_s)$. In the second step, we estimate the transport map $\vartheta_{n,:}$ via Equation (5) using the remaining samples. As shown in the next corollary, Algorithm 1 achieves the desired properties.

**Corollary 4.2.** *Under the same conditions as in Theorem 3.4, Algorithm 1 achieves the upper bound in Theorem 3.4 and is consistently fair.*

*Remark* 4.3 (Limitation). The minimization over congruent potentials may present computational challenges. Korotin et al. (2020) proposed adding a penalty term to enforce congruency instead of handling the constraint directly, which may be more practical. Analysis of convergence rates under approximate satisfaction of congruency remains an important direction for future work.

**Connection between fair regression and transport maps estimation in Wasserstein barycenter** To prove Corollary 4.2, we relate the regression error and unfairness of

Algorithm 1 to the estimation error of the estimated transport maps $\vartheta_{n,:}$. Specifically, we demonstrate the connection of Algorithm 1's error and unfairness with $d_{\mu_{\hat{f},:}}^2(\vartheta_{n,:}, \vartheta_{\mu_{\hat{f}}})$ through the following propositions:

**Proposition 4.4.** *Let $\bar{f}_{n,:}$ be a regressor obtained by Algorithm 1. Under the same conditions as in Theorem 3.4, there exists a universal constant $C > 0$ such that*

$$\mathbb{E}_{\mu_:^{2n}}\left[d_{\mu_{X,:}}^2(\bar{f}_{n,:}, \bar{f}_{\mu,:}^*)\right] \leq$$

$$C\left(L^2 \sum_{s \in [M]} w_s \mathcal{E}_{n_s}(\mathcal{P}_s) + \mathbb{E}_{\mu_{\hat{f},:}^n}\left[d_{\mu_{\hat{f},:}}^2(\vartheta_{n,:}, \vartheta_{\mu_{\hat{f},:}}^*)\right]\right),$$

*where $\mathbb{E}_{\mu_{\hat{f},:}^n}$ denotes the expectation over the samples used for constructing $\vartheta_{n,:}$.*

**Proposition 4.5.** *Let $\bar{f}_{n,:}$ be a regressor obtained by Algorithm 1. Under the same conditions as in Theorem 3.4, we have*

$$\inf_{\nu} \max_{s \in [M]} W_2(\bar{f}_{n,s}\sharp\mu_{X,s}, \nu) \leq$$

$$\sqrt{\frac{1}{Mw_{\min}}}d_{\mu_{\hat{f},:}}\left(\vartheta_{n,:}, \vartheta_{\mu_{\hat{f}}}^*\right) \ a.s.,$$

*where $w_{\min} = \min_{s \in [M]} w_s$.*

By Proposition 4.4, we obtain an upper bound on the regression error by deriving an upper bound on $\mathbb{E}_{\mu_:^n}[d_{\mu_{\hat{f},:}}^2(\vartheta_{n,:}, \vartheta_{\mu_{\hat{f},:}}^*)]$. Additionally, Proposition 4.5 implies that Algorithm 1 achieves fairness consistency in Definition 2.2 if $\mathbb{P}_{\mu_:^n}\left\{d_{\mu_{\hat{f},:}}(\vartheta_{n,:}, \vartheta_{\mu_{\hat{f},:}}^*) = o(1)\right\} = 1 - o(1)$. We will provide upper bounds on the error $d_{\mu_{\hat{f},:}}(\vartheta_{n,:}, \vartheta_{\mu_{\hat{f}}})$ through the analyses shown in the next section.

## 5. Transport Maps Estimation in Wasserstein Barycenter

In this section, we investigate the convergence rate of our estimator for transport maps estimation in the Wasserstein barycenter. We conduct our analyses under the general setup of transport maps estimation in the Wasserstein barycenter for real-valued probability measures. We first describe the general setup and demonstrate the convergence rate of our estimator, which is the main result of this section. We then provide detailed analyses to support the convergence rate.

**Setup** Consider the problem of estimating the transport maps that arise in the Wasserstein barycenter problem. Let $\nu_: \in \mathcal{Q}$ be probability measures indexed by $[M]$ on a measurable space $(\Omega, \mathfrak{Z})$. Recall that the Wasserstein barycenter problem involves finding the minimizer of the following optimization problem:

$$\inf_{\nu} \sum_{s \in [M]} w_s W_2^2(\nu_s, \nu).$$

Let $\nu$ be the minimizer of the above optimization problem, i.e., the Wasserstein barycenter. Recall that we denote $\vartheta_{\nu,s}^*$ as the optimal transport map from $\nu_s$ to $\nu$ for each $s \in [M]$. Given $M$ samples each i.i.d. from $\nu_s$, the analyst's goal is to estimate $\vartheta_{\nu,:}^*$. We denote the estimator of $\vartheta_{\nu,:}^*$ as $\vartheta_{n,:}$. According to Propositions 4.4 and 4.5, we assess the error of the estimated transport maps by

$$
\begin{aligned}
d_{\nu_:}\big(\vartheta_{n,:}, \vartheta_{\nu,:}^*\big) &= \sum_{s \in [M]} w_s d_{\nu_s}\big(\vartheta_{n,s}, \vartheta_{\nu,s}^*\big) \\
&= \sum_{s \in [M]} w_s \int \big(\vartheta_{n,s}(z) - \vartheta_{\nu,s}^*(z)\big)^2 \nu_s(dz).
\end{aligned}
$$

Note that $d_{\nu_:}\big(\vartheta_{n,:}, \vartheta_{\nu,:}^*\big)$ is a random variable with randomness stemming from the samples. The goal of the analyses is thus to provide an upper bound on the expectation of the error or a probabilistic upper bound on the error.

**Estimator**  Our estimator $\vartheta_{n,:}$ is constructed by minimizing the empirical multiple correlation over a sieved subset of transport maps. Let $\Theta = \{\vartheta_{\nu,:}^* : \nu_: \in \mathcal{Q}\}$ represent the set of all possible transport maps. Denote by $\nu_{n,:}$ the empirical measures corresponding to $\nu_:$, as determined by the observed samples. Assume that the complexity of $\mathcal{Q}$ is $(\alpha, \beta)$ for some $\alpha > 0$ and $\beta \geq 0$, and let $\{\Theta_j\}_j$ be a sequence of subsets of $\Theta$ as specified in Definition 3.3. The estimator $\vartheta_{n,:}$ is then defined as the solution to the following optimization problem:

$$
\inf_{\vartheta_: \in \Theta_j} C(u_{\vartheta,:}, \nu_{n,:}),
$$

where $j$ is chosen appropriately.

**Estimation Error Bound**  Our main results in this section are expected and probabilistic upper bounds on $d_{\nu_:}(\vartheta_{n,:}, \vartheta_{\nu,:}^*)$.

**Theorem 5.1.** *Let* $\vartheta_{n,:} = \arg\min_{\vartheta_: \in \Theta_j} C(u_{\vartheta,:}; \nu_{n,:})$ *be the estimated transport maps, with $j$ satisfying $2^j \leq \left(\frac{\tilde{n}}{\ln(\tilde{n})}\right)^{1/(\alpha+\beta)} \leq 2^{j+1}$. Suppose that $\vartheta_{\nu,s}^* \in \mathcal{M}_L$ for some $L > 1$, and that $\nu_s$ satisfies the Poincaré-type inequality in Equation (4) for all $\nu_: \in \mathcal{Q}$. Also, suppose that the complexity of $\mathcal{Q}$ is $(\alpha, \beta)$ for some $\alpha > 0$ and $\beta \geq 0$. Then, there exists a constant $C > 0$, possibly depending on $L$ and $M$, such that*

$$
\mathbb{E}_{\nu_:^n}\big[d_{\nu_:}^2(\vartheta_{n,:}, \vartheta_{\nu,:}^*)\big] \leq C\left(\frac{\tilde{n}}{\ln(\tilde{n})}\right)^{-\alpha/(\alpha+\beta)}.
$$

*Moreover, for all $t \geq 1$, with probability at least $1 - 2e^{-t}$,*

$$
d_{\nu_:}^2(\vartheta_{n,:}, \vartheta_{\nu,:}^*) \leq Ct\left(\frac{\tilde{n}}{\ln(\tilde{n})}\right)^{-\alpha/(\alpha+\beta)}.
$$

Theorem 5.1 shows that a larger $\alpha$ and a smaller $\beta$ result in a faster convergence rate.

To establish the upper bound in Theorem 5.1, we follow the convergence rate analysis for the sieved $M$-estimator (see, e.g., Van Der Vaart & Wellner, 2023), since our estimator in Equation (5) can be regarded as a sieved $M$-estimator. The main analytical challenge is that the process $\vartheta_: \to C(u_{\vartheta,:}, \nu_{n,:})$ is not a standard empirical process, and thus standard concentration inequalities do not directly apply. In the following subsection, we present the analysis of $\vartheta_{n,:}$, including a concentration inequality for the process $\vartheta_: \to C(u_{\vartheta,:}, \nu_{n,:})$.

**5.1. Error Analysis for $\vartheta_{n,:}$**

In this subsection, we present the error analysis for $\vartheta_{n,:}$. We begin by formally defining the processes associated with the populational and empirical multiple correlations. The expectation operator $E_\nu$ is defined as $E_\nu u = \mathbb{E}_\nu[u(Z)]$ for a function $u : \mathcal{Z} \to \mathbb{R}$ and $Z \sim \nu$. For a positive integer $n$, the empirical expectation operator $E_{\nu,n}$ is given by $E_{\nu,n}u = \frac{1}{n}\sum_{i=1}^n u(Z_i)$, where $Z_1, ..., Z_n$ are drawn i.i.d. from $\nu$. For measurable functions $u_s : \mathcal{Z} \to \mathbb{R}$ indexed by $[M]$, we define the following operators:

$$
E_{\nu_:} u_: = \sum_{s \in [M]} w_s E_{\nu_s} u_s, \quad E_{n,\nu_:} u_: = \sum_{s \in [M]} w_s E_{n_s,\nu_s} u_s.
$$

The populational and empirical multiple correlations are then expressed as

$$
C(u_{\vartheta,:}, \nu_:) = E_{\nu_:} u_{\vartheta,:}, \quad C(u_{\vartheta,:}, \nu_{n,:}) = E_{n,\nu_:} u_{\vartheta,:}.
$$

**Concentration of the process**  We now establish the concentration inequality for the process $E_{n,\nu_:} - E_{\nu_:}$. We first present a Bernstein-type concentration inequality for fixed potentials $u_{\vartheta,:}$. For $s \in [M]$, let $\nu_s$ be a probability measure on a measurable space $(\mathcal{Z}, \mathfrak{Z})$. Let $Z^{(s)}$ denote a random variable distributed according to $\nu_s$. Let $\mathcal{U}$ be a class of functions $u_s : \mathcal{Z} \to \mathbb{R}$ indexed by $s \in [M]$.

**Proposition 5.2.** *Given* $u_: \in \mathcal{U}$ *such that* $\sum_{s \in [M]} w_s \mathbb{V}_{\nu_s}[u_s(Z^{(s)})] \leq \sigma^2$ *and* $\max_{s \in [M]} |f_s(Z^{(s)}) - E_{\nu_s} u_s| \leq b$ *almost surely, we have*

$$
\mathbb{P}\{(E_{n,\nu_:} - E_{\nu_:})u_: > t\} \leq \exp\left(-\frac{1}{2}\frac{\tilde{n}t^2}{\sigma^2 + tb}\right).
$$

Building on Proposition 5.2, we derive the maximal inequality over the set of functions $\mathcal{U}$. For $u_: \in \mathcal{U}$, define

$$
\sigma_{\nu_:}^2(u_:) = \sum_{s \in [M]} w_s \mathbb{V}_{\nu_s}\big[u_s(Z^{(s)})\big],
$$

$$
b_{\nu_:}(u_:) = \inf_b \left\{ b : \max_{s \in [M]} \left|u_s(Z^{(s)}) - E_{\nu_s} u_s\right| \leq b \text{ a.s.} \right\}.
$$

It is worth noting that both $\sigma_{\nu_:}$ and $b_{\nu_:}$ satisfy the triangle inequality. As complexity measures for $\mathcal{U}$, we introduce Dudley integral-type metrics, defined for $\delta > 0$ as

$$H_{\sigma, \nu_:}(\delta; \mathcal{U}) = \int_0^\delta \sqrt{\ln(N(\epsilon, \mathcal{U}, \sigma_{\nu_:}))} d\epsilon,$$

$$H_{b, \nu_:}(\delta; \mathcal{U}) = \int_0^\delta \ln(N(\epsilon, \mathcal{U}, b_{\nu_:})) d\epsilon.$$

This leads to the following maximal inequality:

**Proposition 5.3.** *Given a fixed $u_:^* \in \mathcal{U}$, $\sigma > 0$, and $b > 0$, define*

$$\mathcal{U}(\sigma, b; u_:^*) = \{u_: \in \mathcal{U} : \sigma_{\nu_:}(u_: - u_:^*) \le \sigma, b_{\nu_:}(u_: - u_:^*) \le b\}.$$

*Then, there exists a universal constant $C > 0$ such that for all $t > 0$,*

$$\mathbb{P}_{\nu_:^n} \left\{ \sup_{u_: \in \mathcal{U}(\sigma, b; u_:^*)} \sqrt{\tilde{n}}(E_{n, \nu_:} - E_{\nu_:})(u_: - u_:^*) \right.$$
$$> C\left( H_{\sigma, \nu_:}(\sigma; \mathcal{U}) + \frac{1}{\sqrt{\tilde{n}}} H_{b, \nu_:}(b; \mathcal{U}) + t \right) \right\}$$
$$\le \exp\left( -\frac{\sqrt{\tilde{n}} t^2}{\sqrt{\tilde{n}}\sigma^2 + tb} \right).$$

**Analyses for Dudley integral-type metrics** To utilize Proposition 5.3, we establish upper bounds on $H_{\sigma, \nu_:}(\delta; \mathcal{U})$ and $H_{b, \nu_:}(\delta; \mathcal{U})$. In the context of analyzing $\vartheta_{n, :, :}$, we provide upper bounds for $\mathcal{U} = \{u_{\vartheta_{\nu_:}^*} : \nu_: \in \mathcal{Q}\}$, where $\mathcal{Q}$ is the set of probability measures with complexity $(\alpha, \beta)$ for some $\alpha > 0$ and $\beta \ge 0$, as specified in Definition 3.3.

**Lemma 5.4.** *Suppose that $\vartheta_{\nu_:}^*$ satisfies the Poincaré-type inequality in Equation (4) for all $\nu_: \in \mathcal{Q}$, and that the complexity of $\mathcal{Q}$ is $(\alpha, \beta)$ for some $\alpha > 0$ and $\beta \ge 0$, as in Definition 3.3. Then, for all $j \ge 0$ and all $\sigma \in (0, \bar{\epsilon}]$ and $\sigma' > 0$,*

$$H_{\sigma, \nu_:}\left(C_P \sigma; \{u_{\vartheta_:} - u_{\vartheta_{\nu_:}^*} : \vartheta_: \in \Theta_j, d_{\nu_:}(\vartheta_:, \vartheta_{\nu_:}^*) \le \sigma'\}\right)$$
$$\le C_P \sqrt{C'} 2^{\beta j / 2} (1 \wedge \sigma \wedge \sigma') \sqrt{1 + \ln(1/(1 \wedge \sigma \wedge \sigma'))},$$

*where $C_P$ is the constant in Equation (4), and $\bar{\epsilon}$ and $C'$ are the constants in Definition 3.3.*

**Lemma 5.5.** *Suppose that $\vartheta_{\nu, s}^* \in \mathcal{M}_L$ for some $L > 1$ and all $\nu_: \in \mathcal{Q}$. Then, there exists a constant $C_b > 0$, possibly depending on $M$, such that for all $b > 0$,*

$$H_{b, \nu_:}(b; \{u_: - u_{\vartheta_{\nu_:}^*} : u_: \in \mathcal{U}, b_{\nu_:}(u_: - u_{\vartheta_{\nu_:}^*}) \le b\})$$
$$\le C_b \sqrt{b}.$$

**Relationship between potentials and transport maps** To apply existing analyses of sieved $M$-estimators to our setting, we establish the relationship between the potential $u_{\vartheta_:}$ and the transport map $\vartheta_:$. Specifically, we prove the following:

**Proposition 5.6.** *Let $\nu_:$ be probability measures indexed by $[M]$ such that $\vartheta_{\nu, :}^* \in \mathcal{M}_L^M$ for some $L > 1$. Then, for all $\vartheta_: \in \mathcal{M}_L^M$ such that $\sum_{s \in [M]} w_s \vartheta_s^{-1}(z) = z$ for all $z \in \Omega$,*

$$\frac{1}{2L} d_{\nu_:}^2(\vartheta_:, \vartheta_{\nu, :}^*) \le E_{\nu_:}\left(u_{\vartheta_:} - u_{\vartheta_{\nu, :}^*}\right) \le \frac{L}{2} d_{\nu_:}^2(\vartheta_:, \vartheta_{\nu, :}^*).$$

**Proof Sketch of Theorem 5.1** The application of Proposition 5.3, together with Lemma 5.4 and Lemma 5.5, demonstrates that for all $k \ge 1$ and $t \ge 1$,

$$\mathbb{P}_{\nu_:^n}\left\{ kt\tau^2 \le d_{\nu_:}^2(\vartheta_{n, :, :}, \vartheta_{j, :, :}^*) \le (k+1)t\tau^2 \right\} \le \exp(-kt),$$

where $\tau^2 = O((\frac{\tilde{n}}{\ln(\tilde{n})})^{-\alpha/(\alpha+\beta)})$, and $\vartheta_{j, :, :}^*$ denotes the minimizer of $C(u_{\vartheta, :, :}, \nu_:)$ over $\vartheta_: \in \Theta_j$. Subsequently, employing the peeling argument in conjunction with Proposition 5.6 leads to the desired results.

# 6. Lower Bound

To establish our lower bound, we develop a technique based on reducing the fair regression estimation problem to a conventional regression estimation problem. Specifically, let $\bar{f}_{n, :}^*$ be the optimal fair regression algorithm satisfying

$$\sup_{\mu_: \in \mathcal{P}} \mathbb{E}_{\mu_:}\left[ d_{\mu_{X, :}}^2\left(\bar{f}_{n, :}^*, \bar{f}_{\mu, :}^*\right) \right] = \bar{\mathcal{E}}_n(\mathcal{P}), \tag{6}$$

We demonstrate that we can construct a conventional regression algorithm using $\bar{f}_{n, :}^*$. The error of this conventional regression algorithm is bounded below by $\mathcal{E}_n(\mathcal{P}_s)$, which consequently provides a lower bound on $\bar{\mathcal{E}}_n(\mathcal{P})$ in terms of $\mathcal{E}_n(\mathcal{P}_s)$.

Consider the scenario where distributions $\mu_1, ..., \mu_M$ are identical, and $\bar{f}_{n, :}^*$ is constructed using samples comprising $n_s$ i.i.d. points from $\mu_s$. Under these conditions, we can construct a regressor for $f_{\mu, 1}^*$ as $f_n = \sum_{s \in [M]} w_s \bar{f}_{n, s}^*$. The error of this regressor provides a lower bound on Equation (6) as follows:

**Theorem 6.1.** *Under the conditions stated in Theorem 3.4, we have*

$$\mathcal{E}_n(\mathcal{P}_1) \le \sup_{\mu_1 \in \mathcal{P}_1: \forall s \in [M], \mu_s = \mu_1} \mathbb{E}_{\mu_:}\left[ d_{\mu_{X, 1}}^2\left(f_n, f_{\mu, 1}^*\right) \right]$$
$$\le \sup_{\mu_: \in \mathcal{P}} \mathbb{E}_{\mu_:}\left[ d_{\mu_{X, :}}^2\left(\bar{f}_{n, :}^*, \bar{f}_{\mu, :}^*\right) \right].$$

This result directly establishes a lower bound on $\bar{\mathcal{E}}_n(\mathcal{P})$ in Theorem 3.4.

## 7. Related Work: Optimal Transport Map Estimation

The problem of optimal transport map estimation in the Wasserstein distance has been extensively studied (Hütter & Rigollet, 2021; Divol et al., 2024; Rigollet & Stromme, 2025; Pooladian et al., 2023; Pooladian & Niles-Weed, 2024; DEB et al., 2021; del Barrio et al., 2023; Manole et al., 2024). The objective of this estimation problem is to estimate the transport map in $W_2(\mu, \mu')$ between two distributions $\mu$ and $\mu'$, given samples from both distributions. Several approaches have been proposed: Manole et al. (2024); DEB et al. (2021); Rigollet & Stromme (2025) utilize plug-in estimators, where they first estimate the joint distribution of $(Z, \vartheta(Z))$ for $Z \sim \mu$ and subsequently construct a transport map by minimizing the expected error between $Z$ and $\vartheta(Z)$. Alternative estimators proposed by Hütter & Rigollet (2021); Divol et al. (2024); del Barrio et al. (2023); Pooladian et al. (2023); Pooladian & Niles-Weed (2024) employ potential minimization techniques. While these studies provide convergence rate analyses for their estimators, their methods cannot be directly applied to optimal transport map estimation in the Wasserstein barycenter problem, as a sample from the barycenter distribution are not observable.

Several researchers have developed methods specifically for optimal transport maps estimation in the Wasserstein barycenter problem, although these methods lack accompanying convergence rate analyses. Korotin et al. (2020) proposed an estimator based on potential minimization, as detailed in Section 2.3. Fan et al. (2021) introduced an estimator utilizing minimax optimization. Korotin et al. (2022) developed an iterative algorithm grounded in the fixed-point theorem established by Álvarez Esteban et al. (2016). Kim et al. (2025) proposed an approach employing a primal-dual coordinate gradient algorithm to estimate the transport maps. Visentin & Cheridito (2025) developed an estimator for the transport maps leveraging the conditional normalizing flow, also demonstrating the application of their estimator to the fair regression problem. While empirical evaluations have shown that these methods achieve low estimation errors, theoretical analyses of their convergence rates remain an open problem.

## 8. Conclusion

We have presented a minimal optimal fair regression algorithm based on post-processing. Our algorithm achieves minimax optimality across various scenarios by leveraging the extensive body of research on minimax optimal conventional regression. Our analysis demonstrates that practitioners can focus their efforts on improving conventional regression methods, which can then be effectively adapted for fair regression applications.

## Acknowledgements

This work was partly supported by JSPS KAKENHI Grant Numbers JP23K13011.

## Impact Statement

This paper presents work whose goal is to advance the field of Machine Learning. There are many potential societal consequences of our work, none which we feel must be specifically highlighted here. Rather, this paper aims at addressing a societal issue of unfairness that may arise in conventional learning algorithms.

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

## A. Missing Proofs

### A.1. Error Analysis for General Sieved $M$-Estimator

In this subsection, we present an error bound for general sieved $M$-estimators. Let $E$ and $E_n$ denote the expectation and empirical process indexed by $\mathcal{U}$, respectively, where $n$ is the sample size. Let $\Theta$ be a parameter space and $\Theta' \subseteq \Theta$ a sieved subset. For a family $u_\theta \in \mathcal{U}$ parameterized by $\theta \in \Theta$, the sieved $M$-estimator is defined as

$$\theta_n = \arg\min_{\theta \in \Theta'} E_n u_\theta.$$

Let $\theta_0 \in \Theta$ denote the ideal parameter such that $E u_{\theta_0} = \inf_{\theta \in \Theta} E u_\theta$, and let $\theta_0' \in \Theta'$ be the ideal parameter within the sieved set, i.e., $E u_{\theta_0'} = \inf_{\theta' \in \Theta'} E u_{\theta'}$. The estimation error of $\theta_n$ is measured by $d(\theta_n, \theta_0)$, where $d$ is a distance function on $\Theta$.

Within this framework, we derive an error bound for sieved $M$-estimators under the following assumptions on the processes $E$ and $E_n$.

**Assumption A.1.** There exist constants $K_{\mathrm{up}}, K_{\mathrm{low}} > 0$ such that for any $\theta \in \Theta$,

$$K_{\mathrm{low}} E(u_\theta - u_{\theta_0}) \leq d^2(\theta, \theta_0) \leq K_{\mathrm{up}} E(u_\theta - u_{\theta_0}).$$

**Assumption A.2.** Let $\gamma \in (1, 2)$ be a constant, and let $a_{0,n}, a_{1,n}, a_{2,n}, b_n > 0$ be sequences of positive numbers associated with $n$. Suppose $H : \mathbb{R} \to \mathbb{R}$ is a non-decreasing function such that for all $t > 0$,

$$\mathbb{P}_{\nu^n} \left\{ \sup_{\theta \in \Theta' : d(\theta, \theta') \leq \sigma} (E_n - E)(u_\theta - u_{\theta'}) > a_{0,n} H(\sigma) + a_{1,n} + a_{2,n} t \right\} \leq \exp\left( -\frac{t^2}{\sigma^2 + b_n t} \right),$$

and $\sigma \to H(\sigma)/\sigma^\gamma$ is non-increasing for $\sigma > 0$.

The following theorem provides the error bound for sieved $M$-estimators.

**Theorem A.3.** *Suppose that* Assumption A.1 *and* Assumption A.2 *hold. Define*

$$\tau_n = \sqrt{4 K_{\mathrm{up}}} \left( \sqrt{2 E(u_{\theta_0'} - u_{\theta_0})} + \sqrt{a_{1,n}} + \sqrt{2 h_n} \right) + \sqrt{4 a_{2,n}^2 + a_{2,n} b_n}, \tag{7}$$

*where $h_n$ is a sequence such that $a_{0,n} H\left( \sqrt{8 K_{\mathrm{up}} h_n} \right) \leq h_n$ for all $n$. Then, for all $t \geq 1$,*

$$\mathbb{P}_{\nu^n} \left\{ d^2(\theta_n, \theta_0) \geq t \left( 2 \tau_n^2 + 2 d^2(\theta_0', \theta_0) \right) \right\} \leq 2 e^{-t}.$$

*Furthermore,*

$$\mathbb{E}_{\nu^n} \left[ d^2(\theta_n, \theta_0) \right] \leq 4 \ln(2) \tau_n^2 + 2 d^2(\theta_0', \theta_0).$$

By establishing Theorem A.3, one can obtain an error bound for a sieved $M$-estimator by verifying Assumption A.1 and Assumption A.2.

### A.1.1. PROOF OF THEOREM A.3

To establish Theorem A.3, we present the following supporting lemma.

**Lemma A.4.** *Assume there exists $\tau > 0$ such that for all $k \geq 1$ and $t \geq 1$,*

$$\mathbb{P}_{\nu^n} \left\{ k t \tau^2 \leq d^2(\hat{\theta}, \theta_0') \leq (k+1) t \tau^2 \right\} \leq \exp(-kt). \tag{8}$$

*Then, for all $t \geq 1$,*

$$\mathbb{P}_{\nu^n} \left\{ d^2(\hat{\theta}, \theta_0) \geq t \left( 2 \tau^2 + 2 d^2(\theta_0', \theta_0) \right) \right\} \leq 2 e^{-t}.$$

*Furthermore,*

$$\mathbb{E}_{\nu^n} \left[ d^2(\hat{\theta}, \theta_0) \right] \leq 4 \ln(2) \tau^2 + 2 d^2(\theta_0', \theta_0).$$

*Proof of Lemma A.4.* Let $t > 0$ and $\tau > 0$. Define

$$\Theta_k(t) = \left\{\theta \in \Theta', kt\tau^2 \leq d^2(\theta, \theta_0') \leq (k+1)t\tau^2\right\}.$$

By employing the peeling argument,

$$\mathbb{P}_{\nu_:^n}\left\{d^2(\hat{\theta}, \theta_0') \geq t\tau^2\right\} \leq \sum_{k=1}^{\infty} \mathbb{P}_{\nu_:^n}\left\{\hat{\theta} \in \Theta_k(t)\right\}. \tag{9}$$

By assumption, for all $k \geq 1$,

$$\mathbb{P}_{\nu_:^n}\left\{\hat{\theta} \in \Theta_k(t)\right\} \leq \exp(-kt). \tag{10}$$

Combining Equations (9) and (10) results in

$$\mathbb{P}_{\nu_:^n}\left\{d^2(\hat{\theta}, \theta_0') \geq t\tau^2\right\} \leq \sum_{k=1}^{\infty} \exp(-kt) = \frac{e^{-t}}{1 - e^{-t}} \leq 2e^{-t},$$

where the last inequality holds for $t \geq \ln(2)$. By the triangle inequality,

$$d^2(\hat{\theta}, \theta_0) \leq 2d^2(\hat{\theta}, \theta_0') + 2d^2(\theta_0', \theta_0). \tag{11}$$

Therefore, for $t \geq \ln(2)$,

$$\mathbb{P}_{\nu_:^n}\left\{d^2(\hat{\theta}, \theta_0) \geq 2t\tau^2 + 2d^2(\theta_0', \theta_0)\right\} \leq 2e^{-t}.$$

For $t \geq 1$, since $2d^2(\theta_0', \theta_0) \leq 2td^2(\theta_0', \theta_0)$, it follows that

$$\mathbb{P}_{\nu_:^n}\left\{d^2(\hat{\theta}, \theta_0) \geq t\left(2\tau^2 + 2d^2(\theta_0', \theta_0)\right)\right\} \leq 2e^{-t}.$$

For the expectation bound, for any $t_0 > 0$,

$$\begin{aligned}
\mathbb{E}_{\nu_:^n}\left[d^2(\hat{\theta}, \theta_0')\right] &= \int_0^{\infty} \mathbb{P}_{\nu_:^n}\left\{d^2(\hat{\theta}, \theta_0) \geq x\right\}dx \\
&\leq \left(t_0 + \int_{t_0}^{\infty} \mathbb{P}_{\nu_:^n}\left\{d^2(\hat{\theta}, \theta_0') \geq x\tau^2\right\}dx\right)\tau^2 \\
&= \left(t_0 + \int_{t_0}^{\infty} \sum_{k=1}^{\infty} \mathbb{P}_{\nu_:^n}\left\{\hat{\theta} \in \Theta_k(x)\right\}dx\right)\tau^2 \\
&\leq \left(t_0 + \int_{t_0}^{\infty} \sum_{k=1}^{\infty} e^{-kx}dx\right)\tau^2 \\
&= \left(t_0 + \sum_{k=1}^{\infty} \frac{e^{-kt_0}}{k}\right)\tau^2 \\
&= \left(t_0 - \ln\left(1 - e^{-t_0}\right)\right)\tau^2.
\end{aligned}$$

Choosing $t_0 = \ln(2)$ yields

$$\mathbb{E}_{\nu_:^n}\left[d^2(\hat{\theta}, \theta_0')\right] \leq 2\ln(2)\tau^2.$$

Again, by Equation (11),

$$\mathbb{E}_{\nu_:^n}\left[d^2(\hat{\theta}, \theta_0)\right] \leq 4\ln(2)\tau^2 + 2d^2(\theta_0', \theta_0).$$

$\square$

The proof of Theorem A.3 is established by confirming the condition of Equation (8), thereby applying Lemma A.4.

*Proof of Theorem A.3.* By assumption and the triangle inequality, for $\theta \in \Theta'$,

$$
\begin{aligned}
d^2(\hat{\theta}, \theta_0') &\leq 2d^2(\hat{\theta}, \theta_0) + 2d^2(\theta_0', \theta_0) \\
&\leq 2K_{\mathrm{up}}E(u_{\hat{\theta}} - u_{\theta_0}) + 4K_{\mathrm{up}}E(u_{\theta_0'} - u_{\theta_0}) \\
&= 2K_{\mathrm{up}}(E - \hat{E})(u_{\hat{\theta}} - u_{\theta_0'}) + 2K_{\mathrm{up}}\hat{E}(u_{\hat{\theta}} - u_{\theta_0'}) + 4K_{\mathrm{up}}E(u_{\theta_0'} - u_{\theta_0}) \\
&\leq 2K_{\mathrm{up}}(E - \hat{E})(u_{\hat{\theta}} - u_{\theta_0'}) + 4K_{\mathrm{up}}E(u_{\theta_0'} - u_{\theta_0}),
\end{aligned}
$$

where the last inequality uses the fact that $\hat{\theta}$ minimizes $\hat{E}u_\theta$ over $\theta \in \Theta'$. For notational convenience, let $U_0 = E(u_{\theta_0'} - u_{\theta_0})$. Then, the left-hand side of Equation (8) can be bounded as

$$
\begin{aligned}
&\mathbb{P}_{\nu_:^n}\left\{ kt\tau^2 \leq d^2(\hat{\theta}, \theta_0') \leq (k+1)t\tau^2 \right\} \\
&\leq \mathbb{P}_{\nu_:^n}\left\{ d^2(\hat{\theta}, \theta_0') \leq (k+1)t\tau^2, (E - \hat{E})(u_{\hat{\theta}} - u_{\theta_0'}) \geq \frac{kt\tau^2}{2K_{\mathrm{up}}} - 2U_0 \right\} \\
&\leq \mathbb{P}_{\nu_:^n}\left\{ \sup_{\theta \in \Theta': d^2(\theta, \theta_0') \leq (k+1)t\tau^2} (E - \hat{E})(u_{\hat{\theta}} - u_{\theta_0'}) \geq \frac{kt\tau^2}{2K_{\mathrm{up}}} - 2U_0 \right\}.
\end{aligned}
\tag{12}
$$

By the concentration inequality assumption and Equation (12), letting

$$
\gamma_{k,t}(\tau) = \frac{\tau}{2K_{\mathrm{up}}a_{2,n}} - \frac{2U_0}{a_{2,n}kt\tau} - \frac{a_{1,n}}{a_{2,n}kt\tau} - \frac{a_{0,n}}{a_{2,n}kt\tau}H(\sqrt{(k+1)t}\tau),
$$

the condition in Equation (8) holds if

$$
\frac{\gamma_{k,t}^2(\tau)}{(1 + \frac{1}{k}) + \frac{b_n}{\tau}\gamma_{k,t}(\tau)} \geq 1,
\tag{13}
$$

provided $\gamma_{k,t}(\tau) > 0$.

We now seek $\tau$ such that Equation (13) and $\gamma_{k,t}(\tau) > 0$ are satisfied. For $c, b > 0$, the function $x \mapsto \frac{x^2}{c+bx}$ is non-decreasing for $x \geq 0$. Thus, Equation (13) holds if a lower bound $\gamma_{k,t}(\tau) \geq \bar{\gamma}(\tau)$ satisfies

$$
\frac{\bar{\gamma}^2(\tau)}{(1 + \frac{1}{k}) + \frac{b_n}{\tau}\bar{\gamma}(\tau)} \geq 1.
$$

Since $\sigma \mapsto H(\sigma)/\sigma^\gamma$ is non-increasing, $H(a\sigma)/(a\sigma)^\gamma \leq H(\sigma)/\sigma^\gamma$ for all $a \geq 1$ and $\tau > 0$. Therefore, for $k \geq 1$ and $t \geq 1$,

$$
\begin{aligned}
\gamma_{k,t}(\tau) &\geq \frac{\tau}{2K_{\mathrm{up}}a_{2,n}} - \frac{2U_0}{a_{2,n}kt\tau} - \frac{a_{1,n}}{a_{2,n}kt\tau} - \frac{a_{0,n}((k+1)t)^{\frac{\gamma}{2}}}{a_{2,n}kt\tau}H(\tau) \\
&\geq \frac{\tau}{2K_{\mathrm{up}}a_{2,n}} - \frac{2U_0}{a_{2,n}\tau} - \frac{a_{1,n}}{a_{2,n}\tau} - \frac{2a_{0,n}}{a_{2,n}\tau}H(\tau) \coloneqq \bar{\gamma}(\tau).
\end{aligned}
$$

We can rewrite $\bar{\gamma}(\tau)$ as

$$
\bar{\gamma}(\tau) = \tau^{\gamma-1}\left( \frac{\tau^{2-\gamma}}{2K_{\mathrm{up}}a_{2,n}} - \frac{2U_0}{a_{2,n}\tau^\gamma} - \frac{a_{1,n}}{a_{2,n}\tau^\gamma} - \frac{2a_{0,n}}{a_{2,n}\tau^\gamma}H(\tau) \right).
$$

Since $\sigma \mapsto H(\sigma)/\sigma^\gamma$ is non-increasing, $\bar{\gamma}(\tau)$ is a product of two non-decreasing functions in $\tau$, and thus is non-decreasing in $\tau$ when these functions are non-negative. Let $\tau^* > 0$ satisfy

$$
\frac{\tau^*}{2K_{\mathrm{up}}} - \frac{2U_0}{\tau^*} - \frac{a_{1,n}}{\tau^*} - \frac{2a_{0,n}}{\tau^*}H(\tau^*) \geq 0.
\tag{14}
$$

Then, for $\tau \geq \tau^*$, $\bar{\gamma}(\tau)$ is non-decreasing in $\tau$, and for any $x > 0$, $\bar{\gamma}(\tau^* + 2K_{\text{up}}a_{2,n}x) \geq x$. Noting that $\bar{\gamma}(\tau)/\tau \leq \frac{1}{2K_{\text{up}}a_{2,n}}$, Equation (13) holds if $\bar{\gamma}^2(\tau) \geq 2 + \frac{b_n}{2K_{\text{up}}a_{2,n}}$. Therefore, for $\tau^*$ satisfying Equation (14), Equation (13) holds with

$$\tau = \tau^* + \sqrt{8K_{\text{up}}^2 a_{2,n}^2 + 2K_{\text{up}}a_{2,n}b_n}.$$

It remains to derive $\tau^*$ satisfying Equation (14). Since $\sigma \mapsto H(\sigma)/\sigma^\gamma$ is non-increasing, for any $\tau' \in (0, \tau^*]$,

$$\frac{H(\tau^*)}{\tau^*} \leq \frac{\tau^* H(\tau')}{\tau'^2}.$$

Define

$$\tau^* = \sqrt{4K_{\text{up}}}\left(\sqrt{2U_0} + \sqrt{a_{1,n}} + \sqrt{2h_n}\right),$$

where $h_n$ is defined in the statement. Then,

$$\frac{\tau^*}{2K_{\text{up}}} - \frac{2U_0}{\tau^*} - \frac{a_{1,n}}{\tau^*} - \frac{2a_{0,n}}{\tau^*}H(\tau^*)$$

$$\geq \sqrt{\frac{4U_0}{2K_{\text{up}}}} + \sqrt{\frac{2a_{1,n}}{2K_{\text{up}}}} + \sqrt{\frac{4h_n}{2K_{\text{up}}}} - \frac{2U_0}{\sqrt{4K_{\text{up}}}\sqrt{2U_0}} - \frac{a_{1,n}}{\sqrt{4K_{\text{up}}}\sqrt{a_{1,n}}}$$

$$- \frac{2a_{0,n}H\left(\sqrt{8K_{\text{up}}h_n}\right)}{8K_{\text{up}}h_n}\sqrt{4K_{\text{up}}}\left(\sqrt{2U_0} + \sqrt{a_{1,n}} + \sqrt{2h_n}\right)$$

$$\geq \left(\sqrt{2} - \frac{2}{\sqrt{2}}\right)\sqrt{\frac{2U_0}{2K_{\text{up}}}} + \left(\sqrt{2} - \frac{2}{\sqrt{2}}\right)\sqrt{\frac{a_{1,n}}{2K_{\text{up}}}} + \left(\sqrt{2} - \frac{1}{\sqrt{2}}\right)\sqrt{\frac{2h_n}{2K_{\text{up}}}} \geq 0,$$

where we use the assumption of $a_{0,n}H\left(\sqrt{8K_{\text{up}}h_n}\right) \leq h_n$ to derive the last inequality. The resulting $\tau$ then coincides with $\tau_n$ as defined in Equation (7), verifying Equation (8) with $\tau_n$. Application of Lemma A.4 completes the proof. $\square$

## A.2. Proofs for Section 5

### A.2.1. PROOF OF THEOREM 5.1

*Proof of Theorem 5.1.* In the context of Appendix A.1, we identify $E = E_{\nu_{\cdot}}$, $E_n = E_{n,\nu_{\cdot}}$, $u_\theta = u_{\vartheta_{\cdot,:}}$, and $d = d_{\nu_{\cdot}}$. Furthermore, we set $\theta_0 = \vartheta_{\nu_{\cdot},:}^*$, $\theta_n = \vartheta_{n,:}$, and $\theta_0' = \vartheta_{j,:}^*$, where $\vartheta_{j,:}^*$ denotes the minimizer of $C(u_{\vartheta_{\cdot,:}}, \nu_{\cdot})$ over $\vartheta_{\cdot} \in \Theta_j$. To apply Theorem A.3, it is necessary to verify Assumption A.1 and Assumption A.2.

By Proposition 5.6, the processes $E = E_{\nu_{\cdot}}$ and $E_n = E_{n,\nu_{\cdot}}$ satisfy Assumption A.1 with $K_{\text{low}} = \frac{1}{2L}$ and $K_{\text{up}} = \frac{L}{2}$. By the Poincaré-type inequality assumption, if $\vartheta_{\cdot} \in \Theta_j$ satisfies $d_{\nu_{\cdot}}(\vartheta_{\cdot}, \vartheta_{\nu_{\cdot},:}^*) \leq \sigma$, then $E_{\nu_{\cdot}}(u_{\vartheta_{\cdot,:}} - u_{\vartheta_{\nu_{\cdot},:}^*}) \leq C_P\sigma$. Since $\Omega$ is bounded, for $\vartheta_{\cdot} \in \Theta_j \subseteq \mathcal{M}_L^M$, there exists a constant $b > 0$ such that $b_{\nu_{\cdot}}(u_{\vartheta_{\cdot,:}} - u_{\vartheta_{\nu_{\cdot},:}^*}) \leq b$, which follows from the smoothness of $u_{\vartheta,s}$ and $u_{\vartheta_{\nu_{\cdot},s}^*}$. Therefore, by applying Proposition 5.3 together with Lemma 5.4 and Lemma 5.5, these processes satisfy Assumption A.2 with

$$a_{0,n} = \frac{CC_P\sqrt{C'}2^{\beta j/2}}{\sqrt{\tilde{n}}}$$

$$a_{1,n} = \frac{CC_b\sqrt{b}}{\tilde{n}}$$

$$a_{2,n} = \frac{C}{\sqrt{\tilde{n}}}$$

$$b_n = \frac{b}{\sqrt{\tilde{n}}}$$

$$H(\sigma) = (1 \wedge \sigma)\sqrt{1 + \ln(1/(1 \wedge \sigma))},$$

where $C$ is the constant in Proposition 5.3. With the choice of $j$ as stated, we have

$$a_{0,n} \leq CC_P \sqrt{C'} \frac{\tilde{n}^{-\alpha/2(\alpha+\beta)}}{\ln^{\beta/2(\alpha+\beta)}(\tilde{n})}.$$

Set $h_n = c(\ln(\tilde{n})/\tilde{n})^{\alpha/(\alpha+\beta)}$ for some $c > 0$ such that $4Lh_n \leq 1$. Then,

$$\frac{a_{0,n} H(\sqrt{4Lh_n})}{h_n} \leq CC_P \sqrt{\frac{4C'L}{c}} \ln^{-1/2}(\tilde{n}) \sqrt{1 + \frac{\alpha}{2(\alpha+\beta)} \ln(\tilde{n}) - \frac{1}{2} \ln\left(4Lc \ln^{\frac{\alpha}{\alpha+\beta}}(\tilde{n})\right)}. \tag{15}$$

The right-hand side of Equation (15) is monotonically decreasing in $c$ and $\tilde{n}$. Hence, for sufficiently large $c$ independent of $\tilde{n}$, we have $\frac{a_{0,n} H(\sqrt{4Lh_n})}{h_n} \leq 1$. Thus, by applying Theorem A.3, we obtain the error bound with

$$\tau_n = \sqrt{2L} \left( \sqrt{2E(u_{\vartheta_j^*,:} - u_{\vartheta_\nu^*,:})} + \sqrt{\frac{CC_b}{\tilde{n}}} + \sqrt{2c\left(\frac{\ln(\tilde{n})}{\tilde{n}}\right)^{\alpha/(\alpha+\beta)}} + \sqrt{\frac{4C^2}{\tilde{n}} + \frac{CC_b}{\tilde{n}}} \right).$$

From Proposition 5.6 and Definition 3.3, it follows that

$$\begin{aligned}
E_{\nu_:}(u_{\vartheta_{j,:}^*} - u_{\vartheta_{\nu,:}^*}) &= \inf_\vartheta E_{\nu_:}(u_{\vartheta_:} - u_{\vartheta_{\nu,:}^*}) \\
&\leq 2L \inf_\vartheta d_{\nu_:}^2(\vartheta_:, \vartheta_{\nu,:}^*) \leq 2LC2^{-\alpha j},
\end{aligned}$$

and

$$d_{\nu_:}^2(\vartheta_{j,:}^*, \vartheta_{\nu,:}^*) \leq \frac{L}{2} E_{\nu_:}(u_{\vartheta_{j,:}^*} - u_{\vartheta_{\nu,:}^*}) \leq 2L^2 C2^{-\alpha j}.$$

With the choice of $j$ as stated, we have

$$2^{-\alpha j} \leq \frac{1}{2^\alpha} \left(\frac{\tilde{n}}{\ln(\tilde{n})}\right)^{-\alpha/(\alpha+\beta)}.$$

Therefore,

$$\begin{aligned}
\tau_n^2 &= O\left(\left(\frac{\tilde{n}}{\ln(\tilde{n})}\right)^{-\alpha/(\alpha+\beta)}\right) \\
d_{\nu_:}^2(\vartheta_{j,:}^*, \vartheta_{\nu,:}^*) &= O\left(\left(\frac{\tilde{n}}{\ln(\tilde{n})}\right)^{-\alpha/(\alpha+\beta)}\right),
\end{aligned}$$

which establishes the claim. □

### A.2.2. PROOF OF LEMMA 5.4 AND LEMMA 5.5

Here, we present the proofs of Lemma 5.4 and Lemma 5.5.

*Proof of Lemma 5.4.* By the Poincaré-type inequality,

$$N\left(\epsilon, \{u_{\vartheta,:} - u_{\vartheta_\nu^*,:} : \vartheta_: \in \Theta_j\}, \sigma_{\nu_:}\right) \leq N\left(\frac{\epsilon}{C_P}, \Theta_j, d_{\nu_:}\right).$$

According to Definition 3.3, we have

$$\ln N\left(\epsilon, \{\vartheta_: : \vartheta_: \in \Theta_j, d_{\nu_:}(\vartheta_:, \vartheta_{\nu,:}^*) \leq \sigma'\}, d_{\nu_:}\right) \leq C' 2^{\beta j} \ln_+(1/\epsilon),$$

for $\epsilon \leq \sigma' \wedge \bar{\epsilon}$, and the log-covering number is zero if $\epsilon > \sigma'$. Therefore,

$$
\begin{aligned}
H_{\sigma,\nu_:}&\big(C_P\sigma; \big\{u_{\vartheta_:,:} - u_{\vartheta^*_{\nu_:,:}} : \vartheta_: \in \Theta_j, d_{\nu_:}(\vartheta_:, \vartheta^*_{\nu_:,:}) \leq \sigma'\big\}\big) \\
&= \int_0^{C_P\sigma} \sqrt{\ln N\big(\epsilon, \big\{u_{\vartheta_:,:} - u_{\vartheta^*_{\nu_:,:}} : \vartheta_: \in \Theta_j, d_{\nu_:}(\vartheta_:, \vartheta^*_{\nu_:,:}) \leq \sigma'\big\}, \sigma_{\nu_:}\big)} d\epsilon \\
&\leq \int_0^{C_P(\sigma\wedge\sigma')} \sqrt{C' 2^{\beta j} \ln_+(C_P/\epsilon)} d\epsilon \\
&\leq C_P \sqrt{C'} 2^{\beta j/2} \int_0^{\sigma\wedge\sigma'} \sqrt{\ln_+(1/\epsilon)} d\epsilon.
\end{aligned}
$$

By the Cauchy-Schwarz inequality, for any $z \in (0,1)$,

$$
\begin{aligned}
\int_0^z \sqrt{\ln(1/\epsilon)} d\epsilon &\leq \sqrt{\int_0^z 1 d\epsilon \int_0^z \ln(1/\epsilon) d\epsilon} \\
&= \sqrt{z(z\ln(1/z) + z)} \leq z\sqrt{1 + \ln(1/z)}.
\end{aligned}
$$

Thus,

$$
H_{\sigma,\nu_:}\big(C_P\sigma; \big\{u_{\vartheta_:,:} - u_{\vartheta^*_{\nu_:,:}} : \vartheta_: \in \Theta_j, d_{\nu_:}(\vartheta_:, \vartheta^*_{\nu_:,:}) \leq \sigma'\big\}\big) \leq C_P\sqrt{C'} 2^{\beta j/2}(1 \wedge \sigma \wedge \sigma')\sqrt{1 + \ln(1/(1 \wedge \sigma \wedge \sigma'))}.
$$

$\square$

*Proof of Lemma 5.5.* Let $\|\cdot\|_\infty$ denote the $\infty$-norm over $\Omega$. For any $\mathcal{U}$, $N(\epsilon, \mathcal{U}, b_{\nu_:}) \leq N_\infty(\epsilon, \mathcal{U}, \|\cdot\|_\infty)$, since $\nu_: \in \mathcal{Q}$ is supported only on $\Omega$. For any $\vartheta_: \in \mathcal{M}_L^M$, $u_{\vartheta_:,s}$ belongs to the Hölder class with smoothness 2. Interpreting $u_{\vartheta_:,:}$ as a vector-valued function, it follows from Park & Muandet (2023) that there exists a constant $C > 0$, possibly depending on $M$, such that $\ln N(\epsilon, \big\{u_: - u_{\vartheta^*_{\nu_:,:}} : u_: \in \mathcal{U}\big\}, \|\cdot\|_\infty) \leq C\epsilon^{-1/2}$. Therefore,

$$
H_{b,\nu_:}\big(b; \big\{u_: - u_{\vartheta^*_{\nu_:,:}} : u_: \in \mathcal{U}, b_{\nu_:}(u_: - u_{\vartheta^*_{\nu_:,:}}) \leq b\big\}\big) \leq \int_0^b C\epsilon^{-1/2} d\epsilon = 2C\sqrt{b},
$$

which establishes the claim. $\square$

### A.2.3. PROOF OF PROPOSITION 5.2

*Proof of Proposition 5.2.* We derive a high probability upper bound through bounding the cumulant generating function. Given a random variable $X$ on a probability space $(\mathcal{Z}, \mathfrak{Z}, \nu)$, define the cumulant generating function of $X$ as for $\lambda \in \mathbb{R}$,

$$
\kappa(\lambda; X) = \ln(\mathbb{E}_\nu[\exp(\lambda X)]).
$$

The Chernoff bound implies that for $\lambda \geq 0$,

$$
\mathbb{P}_\nu\{X \geq t\} \leq \exp(-\lambda t + \kappa(\lambda; X)). \tag{16}
$$

Hence, an upper bound on the cumulant generating function yields the corresponding high probability upper bound.

Let $Z_1^{(s)}, ..., Z_{n_s}^{(s)}$ be $n_s$ i.i.d. copies of $Z^{(s)}$. Let $\sigma_s^2 = \mathbb{V}_{\nu_s}[u_s(Z^{(s)})]$, and let $b_s$ be the minimum real such that $u_s(Z^{(s)}) - \mathbb{E}_{\nu_s}[u_s(Z^{(s)})] \leq b_s$ almost surely. For $\lambda > 0$, we derive an upper bound on the cumulant generating function of $(E_{n,\nu_:} - E_{\nu_:})u_:$. By the independence among $Z_:^{(:)}$, we have

$$
\kappa(\lambda; (E_{n,\nu_:} - E_{\nu_:})u_:) = \sum_{s\in[M]} n_s \ln\bigg(\mathbb{E}_{\nu_s}\bigg[\exp\bigg(\lambda\frac{w_s}{n_s}\bigg(u_s(Z_1^{(s)}) - \int u_s(z)\nu_s(dz)\bigg)\bigg)\bigg]\bigg).
$$

Noting that the Bernstein's condition is satisfied for a bounded random variable, we obtain

$$
\ln\bigg(\mathbb{E}_{\nu_s}\bigg[\exp\bigg(\lambda\frac{w_s}{n_s}\bigg(u_s(Z_1^{(s)}) - \int u_s(z)\nu_s(dz)\bigg)\bigg)\bigg]\bigg) \leq \frac{\big|\lambda\frac{w_s}{n_s}\big|^2 \sigma_s^2}{2\big(1 - b_s\big|\lambda\frac{w_s}{n_s}\big|\big)},
$$

provided that $\left| \lambda \frac{w_s}{n_s} \right| \leq \frac{1}{b_s}$. Hence,

$$\kappa(\lambda; (E_{n,\nu_:} - E_{\nu_:})u_:) \leq \sum_{s \in [M]} n_s \frac{\lambda^2 \frac{w_s^2}{n_s^2} \sigma_s^2}{2 \left( 1 - b_s \left| \lambda \frac{w_s}{n_s} \right| \right)},$$

as long as $\left| \lambda \frac{w_s}{n_s} \right| \leq \frac{1}{b_s}$ for all $s \in [M]$.

Define $\bar{\sigma}^2 = \sum_{s \in [M]} \frac{w_s^2 \sigma_s^2}{n_s} \leq \frac{\sigma^2}{\tilde{n}}$ and $\bar{b} = \max_{s \in [M]} \frac{w_s b_s}{n_s} \leq \frac{b}{\tilde{n}}$. Set $\lambda = \frac{t}{\bar{\sigma}^2 + t\bar{b}}$, which satisfies $0 \leq \lambda \frac{w_s}{n_s} \leq \frac{1}{b_s}$ for all $s \in [M]$. Then, we have

$$\kappa(\lambda; (E_{n,\nu_:} - E_{\nu_:})u_:) - \lambda t$$

$$\leq \sum_{s \in [M]} n_s \frac{t^2 \frac{w_s^2}{n_s^2} \sigma_s^2}{2 \left( \bar{\sigma}^2 + t\bar{b} \right)^2 \left( 1 - b_s \left| \lambda \frac{w_s}{n_s} \right| \right)} - \frac{t^2}{\bar{\sigma}^2 + t\bar{b}}$$

$$= \sum_{s \in [M]} n_s \frac{t^2 \frac{w_s^2}{n_s^2} \sigma_s^2}{2 \left( \bar{\sigma}^2 + t\bar{b} \right) \left( \bar{\sigma}^2 + t\bar{b} - t \frac{w_s b_s}{n_s} \right)} - \frac{t^2}{\bar{\sigma}^2 + t\bar{b}}$$

$$\leq \sum_{s \in [M]} n_s \frac{t^2 \frac{w_s^2}{n_s^2} \sigma_s^2}{2 \left( \bar{\sigma}^2 + t\bar{b} \right) \bar{\sigma}^2} - \frac{t^2}{\bar{\sigma}^2 + t\bar{b}} = -\frac{t^2}{2(\bar{\sigma}^2 + t\bar{b})} \leq -\frac{\tilde{n} t^2}{2(\sigma^2 + tb)}. \quad (17)$$

The claim follows by applying the Chernoff bound in Equation (16) with the upper bound derived in Equation (17). □

### A.2.4. PROOF OF PROPOSITION 5.3

*Proof of Proposition 5.3.* The desired claim is obtained by applying Van Der Vaart & Wellner (Theorem 2.2.19 and Lemma 2.2.15 2023) with Proposition 5.2 and

$$d_2(u_:, u_:') = \sigma_{\nu_:}^2(u_: - u_:') \quad d_1(u_:, u_:') = \frac{1}{\sqrt{\tilde{n}}} b_{\nu_:}(u_: - u_:').$$

□

### A.2.5. PROOF OF PROPOSITION 5.6

*Proof of Proposition 5.6.* Let $\vartheta_: \in \mathcal{M}_L^M$ be arbitrary. We begin by expressing

$$E_{\nu_:} u_{\vartheta,:}$$

$$= \sum_{s \in [M]} w_s \int u_{\vartheta,s}(z) \nu_s(dz)$$

$$= \sum_{s \in [M]} w_s \int \left( u_{\vartheta,s}(z) + u_{\vartheta,s}^\dagger(\vartheta_{\nu,s}^*(z)) \right) \nu_s(dz) - \sum_{s \in [M]} w_s \int u_{\vartheta,s}^\dagger(\vartheta_{\nu,s}^*(z)) \nu_s(dz)$$

$$= \sum_{s \in [M]} w_s \int \left( u_{\vartheta,s}(z) + u_{\vartheta,s}^\dagger(\vartheta_{\nu,s}^*(z)) \right) \nu_s(dz) - \int \sum_{s \in [M]} w_s u_{\vartheta,s}^\dagger(z) \nu(dz),$$

where the last equality follows from the change of variables $\nu_s = \vartheta_{\nu,s}^{*-1} \sharp \nu$. By the assumption that $\sum_{s \in [M]} w_s \vartheta_s^{-1}(z) = z$ for all $z \in \Omega$, we obtain

$$\int \sum_{s \in [M]} w_s u_{\vartheta_s}^\dagger(z) \nu(dz) = \int \sum_{s \in [M]} w_s \int_0^z \vartheta_s^{-1}(x) dx \, \nu(dz)$$

$$= \int \int_0^z \sum_{s \in [M]} w_s \vartheta_s^{-1}(x) dx \nu(dz)$$

$$= \int \int_0^z x dx \nu(dz) = \frac{1}{2} \int z^2 \nu(dz).$$

Therefore, we have

$$\sum_{s \in [M]} w_s \int u_{\vartheta,s}(z) \nu_s(dz)$$

$$= \sum_{s \in [M]} w_s \int \left( u_{\vartheta,s}(z) + u_{\vartheta,s}^\dagger \big( \vartheta_{\nu,s}^*(z) \big) \right) \nu_s(dz) - \frac{1}{2} \int z^2 \nu(dz). \tag{18}$$

Next, for any $\vartheta'_: \in \mathcal{M}_L^M$, we derive upper and lower bounds for $u_{\vartheta,s}(z) + u_{\vartheta,s}^\dagger(\vartheta'_s(z))$ by employing the approach of Hütter & Rigollet (2021). Since $\vartheta_: \in \mathcal{M}_L^M$, the function $u_{\vartheta,:}$ is $L$-smooth and $1/L$-strongly convex. Thus, for all $z, x \in \mathbb{R}$, it holds that

$$\frac{1}{2L} |z - x|^2 \le u_{\vartheta,s}(z) - u_{\vartheta,s}(x) - \vartheta_s(x)(z - x) \le \frac{L}{2} |z - x|^2.$$

We first establish the lower bound. For $z, x \in \mathbb{R}$, define

$$q_x(z) = u_{\vartheta,s}(x) + \vartheta_s(x)(z - x) + \frac{L}{2} |z - x|^2.$$

The convex conjugate of $q_x$ at $\vartheta'_s(x)$ is given by

$$q_x^\dagger(\vartheta'_s(x)) = -u_{\vartheta,s}(x) + x\vartheta'_s(x) + \frac{1}{2L}(\vartheta'_s(x) - \vartheta_s(x))^2.$$

Since for convex functions $f, g$, $f \le g$ implies $f^\dagger \ge g^\dagger$, it follows that for all $z \in \mathbb{R}$ and any $\vartheta_:, \vartheta'_: \in \mathcal{M}_L^M$,

$$u_{\vartheta,s}(z) + u_{\vartheta,s}^\dagger(\vartheta'_s(z)) \ge z\vartheta'_s(z) + \frac{1}{2L}(\vartheta'_s(z) - \vartheta_s(z))^2. \tag{19}$$

The upper bound is obtained analogously by considering

$$q_x(z) = u_{\vartheta,s}(x) + \vartheta(x)(z - x) + \frac{1}{2L} |z - x|^2,$$

leading to

$$u_{\vartheta,s}(z) + u_{\vartheta,s}^\dagger(\vartheta'_s(z)) \le z\vartheta'_s(z) + \frac{L}{2}(\vartheta'_s(z) - \vartheta_s(z))^2. \tag{20}$$

We now combine Equations (18) to (20) to establish the claim. From Equation (18), we have

$$E_{\nu_:} \big( u_{\vartheta,:} - u_{\vartheta_\nu^*,:} \big)$$

$$= \sum_{s \in [M]} w_s \int \left( u_{\vartheta,s}(z) + u_{\vartheta,s}^\dagger \big( \vartheta_{\nu,s}^*(z) \big) - u_{\vartheta^*,s}(z) + u_{\vartheta^*,s}^\dagger \big( \vartheta_{\nu,s}^*(z) \big) \right) \nu_s(dz). \tag{21}$$

Observe that $D u_{\vartheta',s}(z) = \vartheta'_s(z)$, and the equality case of the Young-Fenchel inequality gives

$$u_{\vartheta',s}(z) + u_{\vartheta',s}^\dagger(\vartheta'_s(z)) = z\vartheta'_s(z). \tag{22}$$

Applying Equations (19) and (22) with $\vartheta'_: = \vartheta_{\nu,s}^*$ in Equation (21), we obtain

$$E_{\nu_:} \big( u_{\vartheta,:} - u_{\vartheta_\nu^*,:} \big)$$

$$\geq \sum_{s \in [M]} w_s \int \frac{1}{2L} \big(\vartheta_s(z) - \vartheta_{\nu,s}^*(z)\big)^2 \nu_s(dz)$$

$$= \frac{1}{2L} \sum_{s \in [M]} w_s d_{\nu_s}^2(\vartheta_s, \vartheta_{\nu,s}^*) = \frac{1}{2L} d_{\nu:}^2(\vartheta_{:,}, \vartheta_{\nu,:}^*)$$

Similarly, by using Equation (20) in place of Equation (19), we have

$$E_{\nu:}\big(u_{\vartheta,:} - u_{\vartheta_\nu^*,:}\big) \leq \frac{L}{2} d_{\nu:}^2(\vartheta_{:,}, \vartheta_{\nu,:}^*)$$

$\square$

## A.3. Proofs for Section 4

## A.4. Proof of Corollary 4.2

*Proof of Corollary 4.2.* According to Theorem 5.1 and Proposition 4.5, there exists a decreasing sequence $r_n$ and a constant $C > 0$ such that, for all $t \geq 1$,

$$\mathbb{P}_{\mu_:^n} \left\{ \inf_\mu \max_{s \in [M]} W_2(\bar{f}_{n,s} \sharp \mu_{X,s}, \mu) > Ctr_n \right\} \leq e^{-t}.$$

By choosing $t = r_n^{-1/2}$, it follows that

$$\mathbb{P}_{\mu_:^n} \left\{ \inf_\mu \max_{s \in [M]} W_2(\bar{f}_{n,s} \sharp \mu_{X,s}, \mu) > o(1) \right\} \leq o(1).$$

Therefore, $\inf_\mu \max_{s \in [M]} W_2(\bar{f}_{n,s} \sharp \mu_{X,s}, \mu)$ converges to 0 in probability. In the case where $\inf_\mu \max_{s \in [M]} W_2(\bar{f}_{n,s} \sharp \mu_{X,s}, \mu) = 0$, it holds that $\bar{f}_{n,1} \sharp \mu_{X,1} = \cdots = \bar{f}_{n,M} \sharp \mu_{X,M}$, which satisfies (strict) demographic parity. Consequently, $\bar{f}_{n,:}$ is consistently fair. The error upper bound can be established directly by combining Theorem 5.1 and Proposition 4.4. $\square$

### A.4.1. PROOF OF PROPOSITION 4.4

*Proof of Proposition 4.4.* We begin by applying the triangle inequality, which yields

$$d_{\mu_{X,:}}\big(\bar{f}_{n,:}, \bar{f}_{\mu,:}^*\big) \leq d_{\mu_{X,:}}\Big(\bar{f}_{n,:}, \vartheta_{\mu_{\hat{f}},:}^* \circ f_{n,:}\Big) + d_{\mu_{X,:}}\Big(\vartheta_{\mu_{\hat{f}},:}^* \circ f_{n,:}, \vartheta_{\mu_{\hat{f}},:}^* \circ f_{\mu,:}^*\Big) + d_{\mu_{X,:}}\Big(\vartheta_{\mu_{\hat{f}},:}^* \circ f_{\mu,:}^*, \bar{f}_{\mu,:}^*\Big). \quad (23)$$

The first term in (23) represents the estimation error of the optimal transport map:

$$d_{\mu_{X,:}}^2\Big(\bar{f}_{n,:}, \vartheta_{\mu_{\hat{f}},:}^* \circ f_{n,:}\Big) = \sum_{s \in [M]} w_s \int \Big((\vartheta_{n,s} \circ f_{n,s})(x) - \big(\vartheta_{\mu_{\hat{f}},s}^* \circ f_{n,s}\big)(x)\Big)^2 \mu_{X,s}(dx)$$

$$= \sum_{s \in [M]} w_s \int \Big(\vartheta_{n,s}(z) - \vartheta_{\mu_{\hat{f}},s}^*(z)\Big)^2 \mu_{\hat{f},s}(dz) = d_{\mu_{\hat{f},:}}^2(\vartheta_{n,:}, \vartheta_{\mu_{\hat{f}},:}^*).$$

For the second term in (23), we establish the $L$-Lipschitz continuity of $\vartheta_{\mu_{\hat{f}},:}^*$. By the first assumption in Assumption 3.1, $f_{n,:} \in \mathcal{F}$ implies $f_{n,:} \sharp \mu_{X,:} \in \mathcal{P}_f$, which ensures that $\vartheta_{\mu_{\hat{f}},:}^* \in \Theta$. The $L$-Lipschitz continuity of $\vartheta_{\mu_{\hat{f}},:}^*$ then follows from Assumption 3.2. Therefore, we can bound the second term as

$$\mathbb{E}_{\mu_:^{2n}}\Big[d_{\mu_{X,:}}^2\Big(\vartheta_{\mu_{\hat{f}},:}^* \circ f_{n,:}, \vartheta_{\mu_{\hat{f}},:}^* \circ f_{\mu,:}^*\Big)\Big] \leq L^2 \mathbb{E}_{\mu_:^n}\Big[d_{\mu_{X,:}}^2\big(f_{n,:}, f_{\mu,:}^*\big)\Big] \leq L^2 \sum_{s \in [M]} w_s \mathcal{E}_{n_s}(\mathcal{P}_s).$$

For the third term in (23), we utilize Proposition 5.6 to obtain

$$d_{\mu_{X,:}}^2\Big(\vartheta_{\mu_{\hat{f}},:}^* \circ f_{\mu,:}^*, \bar{f}_{\mu,:}^*\Big) = d_{\mu_{f,:}}^2\Big(\vartheta_{\mu_{\hat{f}},:}^*, \vartheta_{\mu_f,:}^*\Big)$$

$$\leq 2L E_{\mu_{f,:}} \left( u_{\vartheta^*_{\mu_{\hat{f}}},:} - u_{\vartheta^*_{\mu_f},:} \right)$$

$$\leq 2L \left( E_{\mu_{f,:}} - E_{\mu_{\hat{f}},:} \right) \left( u_{\vartheta^*_{\mu_{\hat{f}}},:} - u_{\vartheta^*_{\mu_f},:} \right)$$

$$\leq 2L \left| \left( E_{\mu_{f,:}} - E_{\mu_{\hat{f}},:} \right) u_{\vartheta^*_{\mu_{\hat{f}}},:} \right| + 2L \left| \left( E_{\mu_{f,:}} - E_{\mu_{\hat{f}},:} \right) u_{\vartheta^*_{\mu_f},:} \right|$$

Let $\varphi_:$ be mappings such that $\varphi_s \sharp \mu_{\hat{f},s} = \mu_{f,s}$. For $\vartheta_: = \vartheta^*_{\mu_{\hat{f}},:}$ or $\vartheta^*_{\mu_f,:}$, we have

$$\left| \left( E_{\mu_{f,s}} - E_{\mu_{\hat{f}},s} \right) u_{\vartheta,s} \right| = \left| \int (u_{\vartheta,s}(\varphi_s(z)) - u_{\vartheta,s}(z)) \mu_{\hat{f},s}(dz) \right|$$

$$= \left| \int \int_z^{\varphi_s(z)} \vartheta_s(x) dx \, \mu_{\hat{f},s}(dz) \right|$$

$$\leq \int \frac{L}{2} (\varphi_s(z) - z)^2 \nu(dz)$$

$$= \frac{L}{2} W_2^2 \left( \mu_{f,s}, \mu_{\hat{f},s} \right) \leq \frac{L}{2} d_{\mu_:}^2 \left( f_{n,:}, f^*_{\mu,:} \right),$$

where we use the identity $u_{\vartheta,s}(x) - u_{\vartheta,s}(y) = \int_y^x \vartheta_s(z) dz$ for all $x, y \in \mathbb{R}$, which follows from Taylor's theorem, to obtain the second equality. Taking expectations, we obtain

$$\mathbb{E}_{\mu_:^{2n}} \left[ d_{\mu_{X,:}}^2 \left( \vartheta^*_{\mu_{\hat{f}},:} \circ f^*_{\mu,:}, \bar{f}^*_{\mu,:} \right) \right] \leq 2L^2 \sum_{s \in [M]} w_s \mathbb{E}_{\mu_:^n} [\mathcal{E}_{n_s}(\mathcal{P}_s)] \leq 2L^2 \sum_{s \in [M]} w_s \mathcal{E}_{n_s}(\mathcal{P}_s).$$

By combining the above results, we have

$$\mathbb{E}_{\mu_:^{2n}} \left[ d_{\mu_{X,:}}^2 \left( \bar{f}_{n,:}, \bar{f}^*_{\mu,:} \right) \right] \leq 3 \mathbb{E}_{\mu_:^{2n}} \left[ d_{\mu_{\hat{f}},:}^2 \left( \vartheta_{n,:}, \vartheta^*_{\mu_{\hat{f}},:} \right) \right] + 9L^2 \sum_{s \in [M]} w_s \mathcal{E}_{n_s}(\mathcal{P}_s),$$

where Jensen's inequality is applied. Finally, due to the mutual independence between $f_{n,:}$ and $\vartheta_{\mu_{\hat{f}},:}$ resulting from sample splitting, we have $\mathbb{E}_{\mu_:^{2n}} \left[ d_{\mu_{X,:}}^2 \left( \vartheta_{n,:}, \vartheta^*_{\mu_{\hat{f}},:} \right) \right] = \mathbb{E}_{\mu_{\hat{f},:}^n} \left[ d_{\mu_{\hat{f}},:}^2 \left( \vartheta_{n,:}, \vartheta^*_{\mu_{\hat{f}},:} \right) \right]$. This completes the proof. $\qquad\square$

### A.4.2. PROOF OF PROPOSITION 4.5

*Proof of Proposition 4.5.* Let $\mu_{\hat{f}}$ be the barycenter of $\mu_{\hat{f},:}$ with weight $w_:$. For any probability measure $\nu$, we have

$$W_2 \left( \bar{f}_{n,s} \sharp \mu_{X,s}, \nu \right) = W_2 \left( \vartheta_{n,s} \sharp \mu_{\hat{f},s}, \nu \right) \leq W_2 \left( \vartheta_{n,s} \sharp \mu_{\hat{f},s}, \vartheta^*_{\mu_{\hat{f}},s} \sharp \mu_{\hat{f},s} \right) + W_2 \left( \vartheta^*_{\mu_{\hat{f}},s} \sharp \mu_{\hat{f},s}, \nu \right)$$

$$\leq d_{\mu_{\hat{f},s}} \left( \vartheta_{n,s}, \vartheta^*_{\mu_{\hat{f}},s} \right) + W_2 \left( \mu_{\hat{f}}, \nu \right).$$

Therefore, we obtain

$$\max_{s \in [M]} W_2 \left( \bar{f}_{n,s} \sharp \mu_{X,s}, \nu \right) \leq \max_{s \in [M]} d_{\mu_{\hat{f},s}} \left( \vartheta_{n,s}, \vartheta^*_{\mu_{\hat{f}},s} \right) + W_2 \left( \mu_{\hat{f}}, \nu \right)$$

$$\leq \sqrt{\frac{1}{M w_{\min}}} d_{\mu_{\hat{f},:}} \left( \vartheta_{n,:}, \vartheta^*_{\mu_{\hat{f}},:} \right) + W_2 \left( \mu_{\hat{f}}, \nu \right).$$

Consequently, it follows that

$$\inf_\nu \max_{s \in [M]} W_2 \left( \bar{f}_{n,s} \sharp \mu_{X,s}, \nu \right) \leq \sqrt{\frac{1}{M w_{\min}}} d_{\mu_{\hat{f},:}} \left( \vartheta_{n,:}, \vartheta^*_{\mu_{\hat{f}},:} \right) + \inf_\nu W_2 \left( \mu_{\hat{f}}, \nu \right) = \sqrt{\frac{1}{M w_{\min}}} d_{\mu_{\hat{f},:}} \left( \vartheta_{n,:}, \vartheta^*_{\mu_{\hat{f}},:} \right).$$

$$\square$$

## A.5. Proofs for Section 6

*Proof of Theorem 6.1.* We begin by noting that, by definition, $\bar{f}_{n,s} \in \mathcal{F}_s$. According to the second requirement in Assumption 3.1, it holds that $\mathcal{F}_1 = \cdots = \mathcal{F}_M$. Furthermore, the third requirement in Assumption 3.1 ensures that $f_n \in \mathcal{F}_1$. Therefore, $f_n$ can be interpreted as the standard regression estimator for $\mathcal{P}_1$ based on a sample of size $n$. By the definition of the minimax error, we have

$$\sup_{\mu_1 \in \mathcal{P}_1 : \forall s \in [M], \mu_s = \mu_1} \mathbb{E}_{\mu_:^n} \left[ d_{\mu_{X,1}}^2 \left( f_n, f_{\mu,1}^* \right) \right] \geq \mathcal{E}_n(\mathcal{P}_1).$$

Next, utilizing the convexity of $d_{\mu_1}^2$, we obtain

$$\sup_{\mu_1 \in \mathcal{P}_1 : \forall s \in [M], \mu_s = \mu_1} \mathbb{E}_{\mu_:^n} \left[ d_{\mu_{X,1}}^2 \left( f_n, f_{\mu,1}^* \right) \right] \leq \sup_{\mu_1 \in \mathcal{P}_1 : \forall s \in [M], \mu_s = \mu_1} \sum_{s \in [M]} w_s \mathbb{E}_{\mu_:^n} \left[ d_{\mu_{X,1}}^2 \left( \bar{f}_{n,s}^*, f_{\mu,1}^* \right) \right]$$

$$= \sup_{\mu_1 \in \mathcal{P}_1 : \forall s \in [M], \mu_s = \mu_1} \sum_{s \in [M]} w_s \mathbb{E}_{\mu_:^n} \left[ d_{\mu_{X,s}}^2 \left( \bar{f}_{n,s}^*, f_{\mu,s}^* \right) \right].$$

When $\mu_1 = \cdots = \mu_M$, the transport map $\vartheta_{\mu,s}^*$ becomes the identity function for all $s$. Thus,

$$\sup_{\mu_1 \in \mathcal{P}_1 : \forall s \in [M], \mu_s = \mu_1} \sum_{s \in [M]} w_s \mathbb{E}_{\mu_:^n} \left[ d_{\mu_{X,s}}^2 \left( \bar{f}_{n,s}^*, f_{\mu,s}^* \right) \right] = \sup_{\mu_1 \in \mathcal{P}_1 : \forall s \in [M], \mu_s = \mu_1} \sum_{s \in [M]} w_s \mathbb{E}_{\mu_:^n} \left[ d_{\mu_{X,s}}^2 \left( \bar{f}_{n,s}^*, \bar{f}_{\mu,s}^* \right) \right]$$

$$= \sup_{\mu_1 \in \mathcal{P}_1 : \forall s \in [M], \mu_s = \mu_1} \mathbb{E}_{\mu_:^n} \left[ d_{\mu_{X,:}}^2 \left( \bar{f}_{n,:}^*, \bar{f}_{\mu,:}^* \right) \right].$$

Moreover, by the second requirement in Assumption 3.1, for any $\mu \in \mathcal{P}_1$, the tuple $\underbrace{(\mu, \ldots, \mu)}_{M \text{ times}}$ belongs to $\mathcal{P}$. Therefore,

$$\sup_{\mu_1 \in \mathcal{P}_1 : \forall s \in [M], \mu_s = \mu_1} \mathbb{E}_{\mu_:^n} \left[ d_{\mu_{X,:}}^2 \left( \bar{f}_{n,:}^*, \bar{f}_{\mu,:}^* \right) \right] \leq \sup_{\mu_: \in \mathcal{P}} \mathbb{E}_{\mu_:^n} \left[ d_{\mu_{X,:}}^2 \left( \bar{f}_{n,:}^*, \bar{f}_{\mu,:}^* \right) \right].$$

$\square$

