# OpenReview forum: "Meta Optimality for Demographic Parity Constrained Regression via Post-Processing"
_ICML.cc/2025/Conference — ICML 2025 poster_

### Official Review · Reviewer_vUCB · 2025-03-11

**Overall Recommendation:** 3

**Summary:**

This paper considers fair regression with respect to statistical parity under the attribute-aware setting. The main focus/contribution is on obtaining a minimax rate for learning fair regressors:

1. Taking the post-processing perspective/approach to achieving fairness, and leveraging the error decomposition result in prior work, this paper bound the minimax rate for fair regression as (minimax rate of the unconstrained regression problem) + (sample complexity of learning the fair post-processor); the fair post-processor is the optimal transports to the barycenter of the output distributions of the unconstrained regressors.
2. The authors cast the problem of learning the barycenter/optimal transports as optimizing over the congruent potentials.
3. A sample complexity bound for learning the barycenter/optimal transports.

**Claims And Evidence:**

Yes

**Essential References Not Discussed:**

Le Gouic et al.  Projection to Fairness in Statistical Learning.  2020.

**Experimental Designs Or Analyses:**

N/A

**Methods And Evaluation Criteria:**

N/A

**Other Comments Or Suggestions:**

- line 180: incomplete sentence
- line 301: Dodley -> Dudley

**Other Strengths And Weaknesses:**

Weaknesses.

1. Theorem 3.4 states a minimax rate in terms of the minimax rate of the unconstrained regression problem plus the sample complexity of learning the barycenter/optimal transports.  A heuristic interpretation of the result is given in remark 3, which says that if the regression problem is *harder* than that of learning the transports ("Such a
situation may frequently happen"), then the minimax rate would be dominated by that of the regression problem.  Unfortunately, no concrete examples is provided to show when would the minimax rate be dominated by the first term.

	In particular, the reviewer is curious about the complexity of learning the barycenter/optimal transport as conveyed in the second term.  For example, on synthetic problems constructed in prior work that analyzes minimax optimality of fair regression, what does the constants $\alpha$ and $\beta$ instantiate to, and does the second term decay faster than the first?

2. Because Theorem 3.4 is derived from the *post-processing* perspective, the second term could have been replaced by any sample complexity bound for learning the barycenter/optimal transports.  How does the derived bound compare (or why incomparable) to that in section 4 of [Chzhen et al., 2020], which was derived for a nonparametric estimate?

3. The result in Theorem 3.4 is obtained by taking the *post-processing* perspective, rather than analyzing the fair regression problem in an *end-to-end* manner (i.e., in-processing), so the caveats of post-processing vs. in-processing applies.  In particular, post-processing can be suboptimal if the class of regressor is constrained, that is, constraining the $\inf_{f_n}$ in $\mathcal E_{n_s}$; see [Woodworth et al., 2017].

4. The algorithm in Section 4 basically takes the same form of that in [Chzhen et al., 2020], except that here the barycenter/optimal transports are estimated by optimizing the congruent potentials on the empirical problem.  It is unknown how it compares to the original nonparametric estimator, and there is no experiment results in this paper, so the contributions in this section is unclear.  What are the benefits of using the new formulation compared to the original?

**Questions For Authors:**

See weaknesses.

**Relation To Broader Scientific Literature:**

- Paper adopts the "post-processing" perspective of [Chzhen et al., 2020] and [Le Gouic et al., 2020] for fair regression, and leveraged its error decomposition to derive the minimax rate (of which there are also prior work, even though are limited to specific data generation processes; see weakness 1).
- The barycenter/optimal transport estimator used in this paper is inspired by [Korotin et al., 2022], for which the authors derived the sample complexity bound.

**Theoretical Claims:**

Did not check proof in detail, but the presented results are reasonable/expected.

---

> ### Author Rebuttal · Authors · 2025-04-01
>
> We appreciate the reviewer’s detailed feedback. Here, we address the main concerns:
>
> W1:
>
> A pertinent example is when $f^*_{\mu,s}$ is a composition of multiple functions, i.e., $f^*_{\mu,s} = g_q \circ ... \circ g_0$, within the Holder class as used by Schmidt-Hieber (2020), and $\vartheta^*_{\mu,s}$ belongs to the Sobolev space. In this scenario, $\mathcal{E}_{n_s}(\mathcal{P}_s) = \Theta(\max_i n_s^{-2\beta^*_i/(2\beta^*_i + t_i)})$, where $\beta^*_i$ represents the cumulative smoothness for $g_q, ..., g_i$, and $t_i$ indicates the dimensionality of the input of $g_i$. For the Sobolev space with smoothness $\gamma > 0$, Definition 3.3 is satisfied with $\alpha = 2\gamma$ and $\beta = 1$ by selecting $\Theta_j$ as a set of functions spanned by the first $j$ wavelet bases. The rate becomes $\max_i n^{-2\beta^*_i/(2\beta^*_i + t_i)} + n^{-2\gamma/(2\gamma + 1)}$, up to logarithmic factors, under the assumption $n_s \ge cw_s n$. If $g_i$ and $\vartheta^* _{\mu,s}$ share the same level of smoothness, i.e., $\beta^* \approx \gamma$, the first term dominates the second if $t_i > 1$ for some $i$. Here, $t_i$ denotes the dimensionality of the essential intermediate data representation. Such an intermediate representation may have multiple dimensions, i.e., $t_i > 1$, causing the first term from the conventional regression problem to dominate the rate.
>
> We will incorporate this example after discussing the implications of the main theorem in the revised version.
>
> W2, W4:
>
> Our results offer significant advantages over the error bounds presented by Chzhen et al. (2020). Firstly, our results are derived under weaker assumptions. Their results necessitate the conventional regression algorithm to have a sub-Gaussian high-probability error bound, whereas our results only require a bound on the expected square error. Additionally, while they demand constant upper and lower bounds on the density of $\nu_i$, we only assume the Poincare-type inequality. This broadens the applicability of our meta-theorem compared to their findings.
>
> Secondly, their results cannot achieve a rate faster than $n^{-1/2}$, as their upper bound on the estimation error of $\vartheta^*_{\mu,:}$ is $n^{-1/2}$ and dominates other terms. However, our results can achieve a rate faster than $n^{-1/2}$ by leveraging the smooth structure of $\vartheta^*_{\mu,:}$.
>
> Thus, our results can demonstrate a faster rate under weaker assumptions compared to those provided by Chzhen et al. (2020). We will include this discussion as an implication of our main theorem in the revised version.
>
> W3:
>
> We wish to clarify that our results do not contradict those of (Woodworth et al., 2017), as their results pertain to equalized odds, not demographic parity. Our findings show that concerning sample size, post-processing can be optimal. However, it may be suboptimal for other parameters, such as the Lipschitz constant $L$ and the number of groups $M$. Analyzing optimality for these parameters is an important direction for our future work.

---

> > ### Comment · Reviewer_vUCB · 2025-04-04
> >
> > The reviewer appreciates the authors' response, and has raised their score!

---

### Official Review · Reviewer_TX3y · 2025-03-12

**Overall Recommendation:** 3

**Summary:**

This paper studied the fair regression problem with demographic parity as a constraint. It claimed that existing minimax optimal regression algorithms are coupled with data generation methods, and proposed meta-theorems to validate the fair minimax optimality. Then they demonstrated that the optimal regression can be achieved through post-processing methods, which thus can be efficiently and flexibly achieved.

**Claims And Evidence:**

The claim that existing analyses are coupled with data generation is not clear. I saw it was mentioned in introduction while I cannot connect it in the main methodology.

**Essential References Not Discussed:**

I have related comments below.

**Experimental Designs Or Analyses:**

No experiments were presented in the paper.

**Methods And Evaluation Criteria:**

This is a theoretical paper which provides analyses instead of new methods.

**Other Comments Or Suggestions:**

Is the proposed post-processing algorithm a new method compared with existing post-processing fairness work? Then would the experiments on it help justify the advances?

**Other Strengths And Weaknesses:**

Strengths: I think the three listed contributions are important if they are correct. The meta-optimality connects the minimax optimal error for fair regression and traditional regression. They demonstrated that optimal regression models can be fair through post-processing method. And they provide convergence rate analysis for the transport map estimation.

Weaknesses:
1. The first thing I felt confused is that in fairness research, minimax optimality also refers to minimizing the risk of the worst group [1], which is under the Rawlsian fairness concept.  I found the authors are working at a different area after I checked the work of Chzhen& Schreuder (2022) and Zeng (2024). The authors mentioned the worst-case error taken over a certain set of data generation models, while I did not understand how this limitation is addressed in this paper.
2. I found regressor f is group-wise (line 75). Then a conventional regressor may be group unaware. How to do post-processing in this case? Maybe I am questioning in an incorrect way.
3. I realised this work and some earlier ones focus on regression analyses. I am curious why regression is considered only? Will these analyses be true for classification as well? A recent paper [2] pointed out loss/error is more sensitive than discrete perditions when considering risk distributions. Is this paper sharing a similar insight?


[1] Minimax Pareto Fairness: A Multi Objective Perspective. ICML 2020.
[2] Towards Harmless Rawlsian Fairness Regardless of Demographic Prior. NeruIPS 2024.

**Questions For Authors:**

I would consider it is a good paper if authors can address my above concerns.

**Relation To Broader Scientific Literature:**

I checked the related papers about OT map estimation in Wasserstein barycenter problems and minimax optimality under regression research. However, I cannot immediately get the corresponding improvements over these works although the authors have highlighted their differences.

**Theoretical Claims:**

I read through the theoretical analyses but only understand some parts of them.

---

> ### Author Rebuttal · Authors · 2025-04-01
>
> We appreciate the reviewer's comprehensive feedback and would like to address their main concerns:
>
> W1: Rawlsian Fairness
>
> We wish to clarify that Rawlsian fairness is fundamentally different from equality-based fairness concepts like demographic parity and equalized odds. Rawlsian fairness focuses on minimizing adverse effects on the most disadvantaged group, whereas equality-based fairness aims for equal treatment across groups. Due to these fundamental differences, results from Rawlsian and equality-based fairness approaches are not directly comparable.
>
> Furthermore, we emphasize that minimax optimality is a well-established concept in statistical literature, used to define the best estimator for a statistical estimation problem. In our context, "minimax" involves maximizing over all possible underlying distributions and minimizing over all estimation algorithms. This concept differs from Rawlsian fairness, as we can define a minimax optimal Rawlsian fair learning algorithm that minimizes the worst-case error, considering both underlying distributions and groups.
>
> Regarding data generation models, they represent potential underlying distributions in nature. Considering a specific set of data generation models is not a limitation of the learner's ability but rather reflects prior knowledge about the data. The sample complexity varies based on this prior knowledge. Extensive research in statistical literature has shown how prior knowledge fundamentally influences sample complexity. Our meta-theorem can accommodate various prior knowledge assumptions by leveraging these existing results.
>
> W2: Group-wise Regressor
>
> We use a group-wise predictor because it is commonly employed in fairness literature. Developing an optimal post-processing method for an unaware predictor is an important area for future research.
>
> W3: Classification
>
> In the context of minimax optimality, regression problems are often considered more fundamental than classification problems, as the minimax optimal error for classification is typically derived using regression techniques. Extending our work to classification problems is a significant direction for future research.

---

### Official Review · Reviewer_pMTi · 2025-03-14

**Overall Recommendation:** 3

**Summary:**

This paper investigates the theoretical properties of fair regression problems by leveraging optimal transport techniques. It provides important theoretical bounds in the context of fair regression, and designs regression algorithm matching the upper bound.

**Claims And Evidence:**

Yes.  The overall structure of this paper is logical and sound, with the logic flow clear.

**Essential References Not Discussed:**

N/A

**Experimental Designs Or Analyses:**

Not applicable

**Methods And Evaluation Criteria:**

Not applicable. This is a theoretical paper.

**Other Comments Or Suggestions:**

N/A

**Other Strengths And Weaknesses:**

N/A

**Questions For Authors:**

N/A

**Relation To Broader Scientific Literature:**

This paper provides new theoretical analysis for the problem of fair regression.

**Theoretical Claims:**

Not in detail - looks reasonable.

---

> ### Author Rebuttal · Authors · 2025-04-01
>
> We thank the reviewer for their positive feedback and for recognizing the importance of our theoretical contributions in fair regression.

---

### Decision · Program_Chairs · 2025-05-01

**Decision:**

Accept (poster)

**Comment:**

This paper addresses the problem of fair regression under demographic parity, with a focus on providing minimax optimality guarantees in an attribute-aware setting. The core contribution is a theoretical framework that decomposes the minimax risk for fair regression into two components: the minimax rate of the unconstrained regression problem and the sample complexity of learning a fair post-processor, which is an optimal transport map to a Wasserstein barycenter. The authors frame this estimation problem through congruent potentials and provide convergence guarantees for the barycenter estimator.

Strengths:

The paper contributes to the theoretical understanding of fair regression by clarifying the statistical cost of post-processing methods relative to unconstrained learning. The meta-theorem and decomposition offer a structured lens for analyzing fair learning under distributional constraints, and extend existing results by [Chzhen et al., 2020] and [Le Gouic et al., 2020]. The authors provide sample complexity bounds for the post-processing component, drawing from recent developments in optimal transport theory (e.g., [Korotin et al., 2022]).

Weaknesses:

Both reviewers noted ambiguity in the claim that existing analyses are tightly coupled with specific data generation processes. Some conceptual issues were raised around the applicability and limitations of post-processing, particularly when the function class of the unconstrained regressor is restricted. There is no direct comparison to related results in [Chzhen et al., 2020] on nonparametric estimation, nor examples illustrating when the regression component dominates the learning cost.

Overall, this is a conceptually valuable and mathematically interesting contribution to the theory of fair regression. It clarifies the statistical limits of fairness through post-processing and advances the literature on minimax optimality in constrained learning. However, the paper would be substantially strengthened by clearer exposition, more thorough justification of certain claims, and concrete examples or experiments that ground the theory.